# DATE-GFN: A Co-Evolutionary Framework for Principled Exploration and Credit Assignment in GFlowNets

## Abstract

Generative Flow Networks (GFlowNets) are powerful for scientific discovery but are severely hampered in sparse-reward, long-horizon settings by the temporal credit assignment problem, which causes high-variance gradients. While recent work has sought to densify learning signals (Jang et al., 2023; Pan et al., 2023a) or improve exploration with methods like Evolution Guided GFlowNets (EGFN) (Ikram et al., 2024c), the fundamental variance issue for the learning agent persists. We introduce the Distillation-Aware Twisted Evolutionary GFlowNet (DATE-GFN), an actor-critic inspired framework that recasts the problem. We advocate for a paradigm shift: instead of evolving policies, DATE-GFN evolves a population of critics (state-dependent value functions, or *twist functions*) that learn to estimate the expected future reward from any state. This constructs a dense, state-dependent guidance signal, transforming the high-variance, reward-driven learning into a stable, low-variance supervised distillation task where the student GFlowNet learns to imitate the policy induced by the best critic. Crucially, we solve the inherent *realization gap* between an optimal teacher and a finite-capacity student via a novel **distillation-aware fitness function**. This objective creates a principled trade-off: it simultaneously rewards critics for discovering high-reward states while penalizing them for their *teachability*, measured by the KL-divergence between their induced policy and the student's. This creates a symbiotic co-evolutionary dynamic where the evolutionary search for better critics is continuously grounded in the student's current learning capabilities. We prove this system converges to a realizable, high-performing equilibrium and show empirically that DATE-GFN significantly outperforms state-of-the-art baselines.

## 1 Introduction

Generative Flow Networks (GFlowNets) have emerged as a principled framework for a central task in scientific discovery: sampling diverse, high-quality candidates $x$ from a vast, structured search space with probability proportional to a reward function, $P(x) \propto R(x)$ (Bengio et al., 2021a). Despite their theoretical elegance, their practical application is severely limited by the problem of *temporal credit assignment*. GFlowNets construct objects via long action trajectories, but the reward $R(x)$ is only delivered at the end. Standard training objectives like Trajectory Balance (TB) (Malkin et al., 2022a) propagate this sparse signal, but the resulting gradient estimators suffer from prohibitively high variance, especially in long-horizon, sparse-reward settings, leading to unstable and inefficient training. This core challenge has spurred several lines of research. One approach is to *densify the learning signal* by decomposing the reward into local potentials (Jang et al., 2023; Pan et al., 2023a). Another is to *improve exploration* to find sparse rewards more often, as exemplified by Evolution Guided GFlowNets (EGFN) (Ikram et al., 2024a), which use an evolutionary algorithm (EA) to discover high-reward trajectories. While valuable, these methods are incomplete. Reward decomposition can be as challenging as the original problem, and exploration-focused methods are palliative, not curative; the GFlowNet agent in EGFN still relies on the high-variance TB objective for learning.

In this paper, we argue for a paradigm shift inspired by actor-critic methods in reinforcement learning (Konda & Tsitsiklis, 2000). Instead of treating the symptoms of high variance, we address the root

cause by fundamentally changing the learning signal itself. We introduce the **Distillation-Aware Twisted Evolutionary GFlowNet (DATE-GFN)**. Our approach is built on a principled decoupling of two distinct challenges: (1) the hard, global exploration problem of discovering the state-value landscape from a sparse signal, and (2) the simpler, local problem of learning a policy to navigate this landscape. DATE-GFN assigns each problem to the right tool. We use an EA not to evolve policies, but to evolve a population of *critics* (or *twist functions*, inspired by SMC (Doucet et al., 2001a; Briers et al., 2010)) that solve the first problem by learning the expected future reward from any state. This transforms the sparse terminal reward into a dense, step-wise learning signal. The GFlowNet policy's task is then reduced to the second problem: a stable, low-variance supervised distillation task to imitate the policy induced by the best critic.

Our central contribution is the mechanism that makes this decoupling robust: a novel **distillation-aware fitness function**. This objective solves the critical **realization gap**—the mismatch between a theoretically optimal teacher and a finite-capacity student—by rewarding critics for both their performance and their *teachability*. This creates a symbiotic co-evolutionary dynamic that grounds the evolutionary search in the student's learning capabilities, guiding the entire system towards a high-performing and, crucially, a *realizable* equilibrium.

## 1.1 OUR CONTRIBUTIONS

This paper makes the following significant contributions:

1. **A Novel Co-Evolutionary GFlowNet Framework.** We introduce DATE-GFN, a new training paradigm that synergistically integrates evolutionary algorithms and GFlowNets. We shift the focus of the evolutionary search from policies to critics (value functions), providing a principled mechanism for solving the temporal credit assignment problem by generating a dense, step-wise reward signal.

2. **The Distillation-Aware Fitness Function for Closing the Realization Gap.** We identify and formalize the *realization gap* as a critical flaw in decoupled teacher-student learning frameworks. Our core technical innovation is the distillation-aware fitness function, which makes the teacher's evolution dependent on the student's learning state. This novel mechanism explicitly optimizes for *teachability* alongside performance, ensuring that the evolved critics are not just powerful but also effectively learnable by a finite-capacity student model. We provided the description of our methodology in Algorithm 1 in Appendix 10.

3. **State-of-the-Art Empirical Performance on Challenging Benchmarks.** We conduct a thorough empirical validation on two difficult domains: the synthetic Hypergrid benchmark and a complex, real-world single-cell perturbation prediction task. Our results demonstrate that DATE-GFN substantially outperforms existing GFlowNet methods, setting a new state-of-the-art for generative modeling in these challenging settings while preserving the crucial ability to generate diverse solutions.

## 2 RELATED WORK

Our work is situated at the intersection of generative flow networks, evolutionary computation, and reinforcement learning, with a central focus on solving the temporal credit assignment problem in sequential generative modeling under sparse rewards.

**Generative Flow Networks and the Credit Assignment Problem.** GFlowNets (Bengio et al., 2021a) are generative models that learn to sample objects in proportion to a reward function $R(x)$. Their primary challenge is temporal credit assignment, as the reward is only observed at the end of a long trajectory. The standard Trajectory Balance (TB) objective (Malkin et al., 2022a) suffers from high-variance gradients in sparse-reward settings (Ikram et al., 2024a). This has motivated a body of work on *densifying the learning signal*. Some approaches leverage known additive reward structures (Pan et al., 2023a) or learn a decomposition of the reward into local potentials (Jang et al., 2023; Mohammadpour et al., 2024). While powerful, these methods can be brittle if the decomposition is hard to learn or relies on specific problem structures. Our work takes an orthogonal approach by learning a value function that directly estimates the total future reward, which is more general.

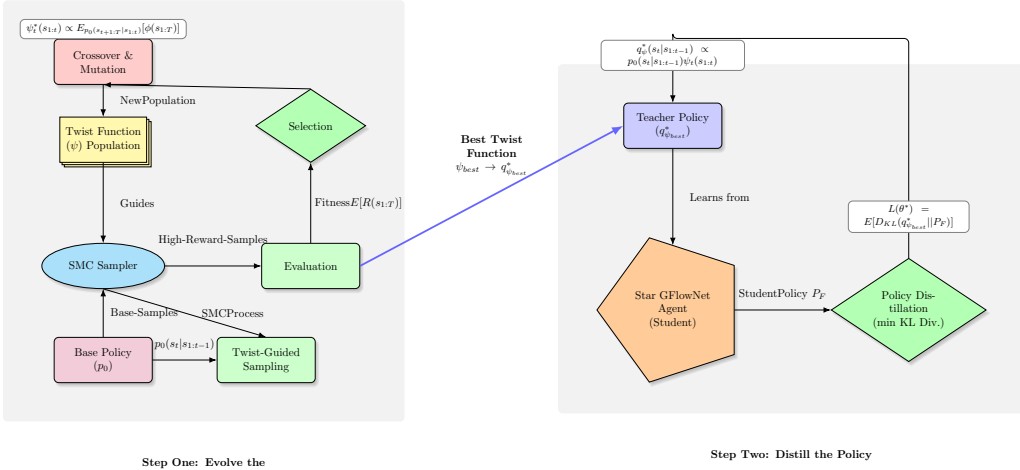

**DATE-GFN Framework**

Figure 1: **Left**: An evolutionary algorithm evolves a population of candidate *twist function* networks (critics) through selection, crossover, and mutation. Their fitness is evaluated by how well they guide an SMC sampler to high-reward outcomes. **Right**: The best critic's induced policy (twist-guided proposal) serves as a teacher for the *star* GFlowNet agent, which is trained to imitate this policy via distillation.

**Evolutionary Algorithms for Policy Search: EGFN.** Evolutionary Algorithms (EAs) are powerful gradient-free methods for exploration (Bäck, 1993; Khadka & Tumer, 2018a). The most relevant predecessor to our work, **EGFN** (Ikram et al., 2024a), evolves a population of GFlowNet policies to enhance exploration. However, EGFN does not solve the underlying credit assignment problem; the final agent is still trained with the high-variance TB objective. It finds *what* trajectories are good but not *why* intermediate steps are valuable. DATE-GFN addresses this gap by evolving value functions (critics) instead of policies, which provides the missing *why* and transforms the learning signal.

**Value Functions in Reinforcement Learning and SMC.** The credit assignment problem is a cornerstone of Reinforcement Learning (RL) (Sutton & Barto, 2018). **Actor-Critic** methods (Konda & Tsitsiklis, 2000) famously solve this by learning a critic (a value function) to provide a dense, low-variance learning signal for the actor (the policy). Our work is also strongly inspired by Sequential Monte Carlo (SMC) methods, where *twist functions* act as learned lookahead value functions to reduce variance and guide sampling (Doucet et al., 2001a; Briers et al., 2010). Recent work has successfully applied this principle to guide generation in large language models (Zhao et al., 2024). DATE-GFN adapts this powerful principle to the GFlowNet context: our *twist function* is an evolved critic, the *student GFlowNet* is the actor, and the distillation-aware co-evolutionary process is the mechanism that allows them to bootstrap each other effectively.

## 3 METHODOLOGY

The proposed framework, DATE-GFN, addresses the critical challenges of credit assignment and exploration in GFlowNets, particularly in settings with sparse rewards and long horizons. This section begins by introducing the foundational concepts from Sequential Monte Carlo (SMC) that motivate our approach. We then critically analyze the limitations of a simple, decoupled framework, identifying the crucial *realization gap*. Finally, we present our advanced, integrated solution: the Distillation-Aware Twisted Evolutionary GFlowNet (DATE-GFN), complete with its theoretical underpinnings and a robust proof of its dynamics.

### 3.1 Twisted SMC: Incorporating a Value Function into Sampling

Before detailing DATE-GFN, we introduce the concept of twist functions from the Sequential Monte Carlo perspective, which provides a formal language for value-guided sampling. In this view, the goal is to sample trajectories $\tau = (s_0, \ldots, s_T)$ from a target distribution $\sigma(\tau)$ proportional to a reward-based potential. Formally, $\sigma(\tau) \propto p_0(\tau)\Phi(\tau)$, where $p_0$ is a simple, tractable base policy (e.g., uniform random) and $\Phi$ is a potential function. For GFlowNet applications, we define this potential to be non-zero only for completed trajectories:

$$\Phi(\tau) = \begin{cases} R(s_T) & \text{if } \tau \text{ is complete (i.e., } s_T \text{ is a terminal state)} \\ 0 & \text{otherwise} \end{cases}$$

Directly sampling from $\sigma$ is intractable. SMC provides a constructive, step-by-step approach guided by *twist functions*. A twist function, $\psi_t(s_{1:t})$, is a non-negative function that provides an intermediate potential at each step of the trajectory. As defined by Briers et al. (2010), the **optimal twist** $\psi_t^*$ is proportional to the exact future cumulative potential, conditioned on the history so far:

$$\psi_t^*(s_{1:t}) \propto \mathbb{E}_{\tau' \sim p_0(\cdot|s_{1:t})}\big[\Phi(\tau_{1:t} \circ \tau')\big] \tag{1}$$

where $\tau'$ represents a trajectory suffix sampled from the base policy. Given our definition of $\Phi$, this has a direct and powerful interpretation in the language of reinforcement learning: the optimal twist function is the *true expected future reward-to-go*, or the state-value function, under the base policy $p_0$:

$$\psi_t^*(s_{1:t}) \propto V^{p_0}(s_t) = \mathbb{E}_{p_0}[R(s_T)|s_t]$$

If this optimal value function $\psi^*$ were known, one could construct a variance-minimal sampling policy, known as the *twist-induced proposal* distribution:

$$q_\psi(s_t \mid s_{1:t-1}) \propto p_0(s_t \mid s_{1:t-1}) \cdot \psi_t(s_{1:t}) \tag{2}$$

This policy, $q_\psi$, can be interpreted as a greedy policy with respect to the value function $\psi$: it biases the selection of the next state $s_t$ towards states that the critic $\psi$ estimates to have higher future value. If the critic is perfect ($\psi = \psi^*$), then $q_\psi$ is the optimal policy that samples trajectories exactly in proportion to $R(x)$, perfectly solving the GFlowNet objective. Of course, $\psi^*$ is unknown, and the central challenge is to learn a good approximation, $\psi_\theta$, for which our framework employs an evolutionary algorithm.

### 3.2 The Challenge of Decoupled Optimization: A Formal Treatment of the Realization Gap

A naive application of the principles above would suggest a simple, two-phase framework:

1. **Phase 1 - Unconstrained Critic Optimization:** Use an Evolutionary Algorithm (EA) to solve for the best possible critic by maximizing expected reward:

$$\theta_\psi^* = \arg\max_{\theta_\psi} \mathbb{E}_{s_{0:T} \sim q_\psi}[R(s_T)]$$

2. **Phase 2 - Policy Distillation:** Freeze the best critic $\psi_{\text{best}} = \psi(\cdot; \theta_\psi^*)$ and train a student GFlowNet policy $P_F(\cdot; \theta^*)$ to mimic its induced policy by minimizing the KL-divergence:

$$\theta_F^{**} = \arg\min_{\theta^*} \mathbb{E}_{s \sim q_{\psi_{\text{best}}}}[D_{KL}(q_{\psi_{\text{best}}}(\cdot|s) \,||\, P_F(\cdot|s; \theta^*))]$$

While this decouples the hard, gradient-free global search from the efficient, gradient-based local policy refinement, it suffers from a critical theoretical flaw: the **realization gap**. The core issue lies in a flawed assumption inherent to the decoupled approach: the EA in Phase 1 operates as if the student GFlowNet were a perfect, infinite-capacity function approximator. It therefore performs an unconstrained search for the best critic in an absolute sense, without any knowledge of the student's finite **representational capacity** and architectural biases (Goodfellow et al., 2016). This creates a fundamental mismatch where the EA, blind to the student's limitations, is free to discover a brilliant critic, $\psi_{\text{genius}}$, whose induced policy $q_{\text{genius}}$ has high complexity (e.g., is non-smooth or has intricate decision boundaries). While theoretically optimal, such a policy may be impossible for the

student's architecture to represent, causing the distillation process to fail and yielding a student with performance far below the teacher's potential. Crucially, this framework has no mechanism to prefer a slightly sub-optimal but "smoother" critic, $\psi_{\text{learnable}}$, whose policy the student could imitate with near-perfect fidelity, even if it would result in a much better final generative model. The optimization objective is thus fundamentally misaligned with the practical goal of producing a high-performing student.

## 3.3 DATE-GFN: A CO-EVOLUTIONARY APPROACH

To bridge this realization gap, we propose the **Distillation-Aware Twisted Evolutionary GFlowNet (DATE-GFN)**, an integrated, co-evolutionary framework where information about the student's learning progress flows back to guide the evolution of the critics.

**Definition 3.1** (Student GFlowNet). The student is a GFlowNet policy parameterized by $\theta^*$:

$$P_F(s_t \mid s_{1:t-1}; \theta^*)$$

**Definition 3.2** (Distillation-Aware Fitness). The fitness of a critic candidate $\psi_j$ with parameters $\theta_{\psi_j}$, evaluated with respect to the *current* state of the student parameters $\theta^*$, is defined as:

$$F_{DA}(\theta_{\psi_j}|\theta^*) = \underbrace{\mathbb{E}_{s_{0:T}\sim q_j}[R(s_T)]}_{\text{Reward Term}} - \lambda \cdot \underbrace{\mathbb{E}_{s_{1:t-1}\sim q_j}[D_{KL}(q_j(\cdot|s_{1:t-1}) \,\|\, P_F(\cdot|s_{1:t-1}; \theta^*))]}_{\text{Teachability Penalty}}, \quad (3)$$

where $q_j$ is the policy induced by $\psi_j$ according to equation 2, and $\lambda \geq 0$ is a hyperparameter balancing reward-seeking and teachability. The training proceeds as a continuous, online loop, as detailed in Algorithm 1 in Appendix 10..

**Remark** The Decoupled Limit: DATE-GFN with $\lambda = 0$. The teachability parameter $\lambda$ connects our co-evolutionary framework to decoupled approaches. When $\lambda = 0$, our fitness function $F_{DA}(\theta_\psi|\theta^*) = \mathcal{R}(\theta_\psi) - \lambda\mathcal{L}(\theta_\psi, \theta^*)$ reduces to $F_{DA}|_{\lambda=0} = \mathcal{R}(\theta_\psi)$. This is precisely the unconstrained, reward-only objective of a decoupled teacher-student framework we term it TE-GFN, where the evolutionary search is blind to the student's capabilities. Thus, the $\lambda = 0$ setting serves as a perfect experimental baseline, allowing us to directly quantify the performance gains from our core contribution—the teachability constraint—and empirically validate the necessity of solving the constrained, realizable optimization problem to close the realization gap.

## 3.4 THEORETICAL ANALYSIS OF THE CO-EVOLUTIONARY DYNAMICS

Our theoretical analysis proceeds in two parts. First, we analyze the dynamics of the practical DATE-GFN algorithm and the properties of its equilibrium state. Second, we provide the mathematical foundation that links this dynamic to our experimental validation.

### 3.4.1 THE DISTILLATION OBJECTIVE AND ITS ROLE

The student's update within the DATE-GFN loop is governed by the distillation loss:

$$\mathcal{L}_{\text{distill}}(\theta^*) = D_{KL}(q_{best}(\cdot|s_{1:t'-1}) \,\|\, P_F(\cdot|s_{1:t'-1}; \theta^*))$$

This is the **Distillation Phase of the co-evolutionary loop**. It is a local, supervised learning problem where the student receives a dense target distribution $q_{best}$ from the teacher critic. The critic effectively "pre-computes" the credit assignment, and the student's task is simply to learn this dense, low-variance signal.

**Proposition 3.3** (Co-evolution towards a Realizable Optimum). *Let the DATE-GFN training process be a dynamical system on the joint parameter space $(\Theta_\psi, \Theta^*)$, where $\Theta_\psi = \{\theta_{\psi_1}, \ldots, \theta_{\psi_k}\}$ and $\Theta^*$ is the space of student parameters. The system's update rules incentivize convergence toward a fixed-point equilibrium $(\Theta_\psi^{**}, \theta^{**})$ characterized by:*

1. ***Student-Teacher Alignment:*** *The student policy $P_F(\cdot; \theta^{**})$ is a close approximation of the policy $q_{best}^{**}$ induced by the best critic in the equilibrium population, $\psi_{best}^{**} \in \{\psi(\cdot; \theta_{\psi_j}^{**})\}$. That is, $D_{KL}(q_{best}^{**} \,\|\, P_F(\cdot; \theta^{**})) \approx 0$.*

2. ***Constrained Optimality of the Teacher:*** *The best critic $\psi_{best}^{**}$ is a member of a high-reward critic population whose induced policies are all structurally representable by the student architecture. It is a solution to the constrained optimization problem of finding a high-reward critic within the set of "teachable" critics.*

*This equilibrium thus represents a high-performing and realizable solution, closing the realization gap by design.*

*Proof Sketch.* The proof analyzes the fixed points of the system's two operators: the Evolutionary Operator ($T_E$) and the Distillation Operator ($T_D$). A fixed point for the student implies $P_F \approx q_{best}$. A fixed point for the EA implies that the best critic, $\psi_{best}$, is a local maximizer of the fitness function $F_{DA}(\cdot|\theta^{**})$. At this point, the teachability penalty for $\psi_{best}$ itself is near zero. Any other critic that is not selected must have a lower fitness, meaning any potential reward gain must be offset by a large teachability penalty. Thus, the equilibrium is a self-consistent state where the teacher is optimal for the student that has learned from it. A detailed proof is provided in Appendix 6. □

### 3.4.2 THE DISTILL-AWARE FITNESS FUNCTION

**Optimal Regime Dynamics and its Constraints.** The parameter $\lambda$ induces three distinct regimes. The *Under-Constrained Regime* ($\lambda \to 0$) leads to fitness functions dominated by reward, $F_{DA} \approx \mathcal{R}(\theta_c)$. The *Over-Constrained Regime* ($\lambda \to \infty$) is dominated by the teachability penalty, $\mathcal{P}(\theta_c|\theta^*) = \lambda\mathbb{E}[D_{KL}(q_j\|P_F)]$, forcing conservative behavior. The *Optimal Balance Regime* (e.g., $\lambda \approx 0.1$) creates a balanced fitness landscape enabling both exploration and stability. This three-regime structure provides a clear framework for understanding our ablation studies validated in 4.1.

**The Mode Escape Condition.** The DATE-GFN framework has an inherent mechanism to resist mode collapse. Consider a dominant critic mode $\psi_1$ to which the student has adapted, making its teachability penalty minimal, $\mathcal{P}(\psi_1|\theta^*) \approx 0$, and its fitness $F_{DA}(\psi_1|\theta^*) \approx \mathcal{R}(\psi_1)$. For a new critic mode $\psi_2$ to be selected, its fitness must be higher:

$$F_{DA}(\psi_2|\theta^*) = \mathcal{R}(\psi_2) - \mathcal{P}(\psi_2|\theta^*) > \mathcal{R}(\psi_1) \quad \text{yields escape condition } \mathcal{R}(\psi_2) - \mathcal{R}(\psi_1) > \lambda\mathcal{L}(\psi_2, \theta^*).$$

This relationship reveals that the population will jump to a new mode if the gain in reward is sufficient to overcome the *cost of teaching* this new strategy to the currently specialized student. This creates a dynamic pressure for diversity, which we validate empirically.

## 4 EXPERIMENTS

To rigorously validate the theoretical claims of DATE-GFN, we selected three complementary experimental domains. The first, the *Hypergrid benchmark*, serves as a canonical, controlled environment to systematically dissect the framework's performance under precisely tunable conditions of reward sparsity and horizon length. The second, the *Antibody Sequence Generation task*, serves as a primary real-world validation, testing the framework's ability to scale to high-dimensional, noisy, and biologically complex problems. The third, the *sEH binder generation task*, pushes the limits of scalability by testing the framework on complex, graph-structured molecular data with state-of-the-art architectures. Together, these tasks provide a comprehensive evaluation of DATE-GFN's capabilities. The implementation details for all experiments are described in Appendix 11.

### 4.1 HYPERGRID EXPERIMENT: A CONTROLLED TESTBED FOR CREDIT ASSIGNMENT.

The Hypergrid environment is designed to isolate and amplify the core challenges that motivate DATE-GFN's architecture: long-horizon credit assignment and extreme reward sparsity. The environment is a $D$-dimensional discrete grid world of side length $H$. The reward function is constructed to create an exponentially sparse landscape with $2^D$ isolated, high-reward modes. In our most challenging setting ($H = 30, D = 5, R_0 = 10^{-5}$), the reward differential spans five orders of magnitude. This setup is explicitly designed to induce the high-variance gradient problem that plagues naive GFlowNet objectives, thereby providing a clear and quantifiable test of whether DATE-GFN's theoretical mechanisms for variance reduction and credit assignment translate into robust performance under pathological conditions. Figure 2 provides compelling evidence for DATE-GFN's systematic superiority across baseline methods.

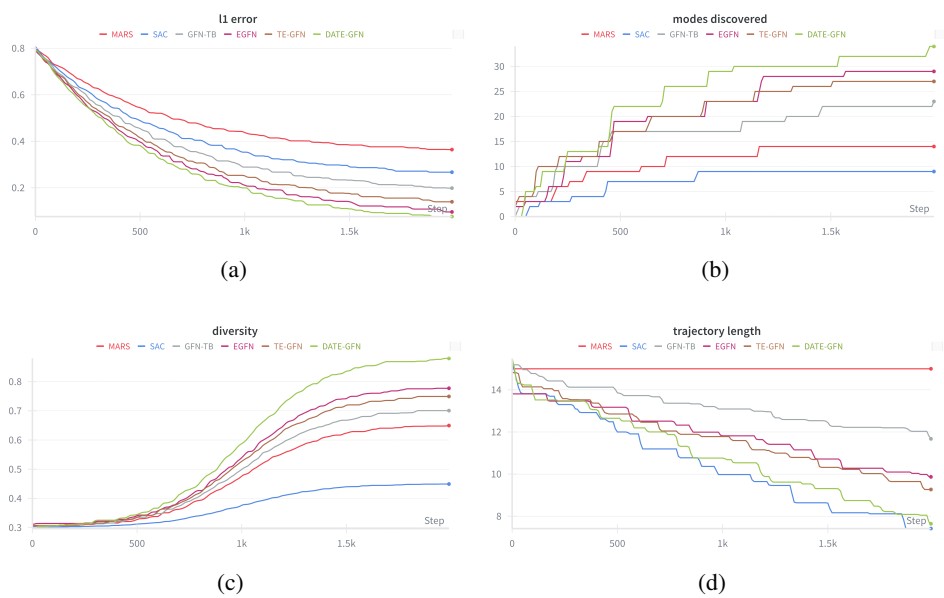

Figure 2: **(Hypergrid) Comparative performance analysis across key metrics.** (a) Relative L1 error showing DATE-GFN's superior convergence to the target distribution. (b) Mode discovery curves demonstrating consistent exploration advantages. (c) Diversity preservation throughout training, maintaining broad sampling coverage. (d) Trajectory length distributions revealing efficient path-finding behavior that balances exploration with exploitation.

**Comparative Performance.** Our results, summarized in Table 1, confirm DATE-GFN's superior performance. We observe a substantial improvement in relative $\ell_1$ error and increase in modes discovered over standard GFN baselines. This demonstrates that DATE-GFN's ability to learn a dense value function via the critic enables it to solve the credit assignment problem where trajectory-level objectives fail. The learned twist function $\psi(s)$ effectively provides intermediate rewards by assigning high value to prefix states that are on a promising path, guiding the student policy through the vast, uninformative regions of the state space.

Table 1: **(Hypergrid) Performance comparison (hardest setting).** Performance in 5-dimensional hypergrid with horizon $H = 30$ and extreme reward sparsity ($R_0 = 10^{-5}$). Mean ± s.e.m. over 8 seeds. (see Appendix 11.2 for Metrics description and 11.3 for task details).

| Method | Rel. $\ell_1 \downarrow$ | Modes $\uparrow$ | Mode Eff. $\uparrow$ | Diversity $\uparrow$ |
|---|---|---|---|---|
| GFN (TB) | $0.62 \pm 0.05$ | $12.1 \pm 2.9$ | $0.94 \pm 0.08$ | $0.41 \pm 0.02$ |
| EGFN | $0.31 \pm 0.04$ | $22.4 \pm 2.1$ | $1.18 \pm 0.06$ | $0.56 \pm 0.02$ |
| TE-GFN | $0.18 \pm 0.03$ | $26.8 \pm 1.4$ | $1.31 \pm 0.05$ | $0.61 \pm 0.02$ |
| **DATE-GFN (ours)** | $\mathbf{0.05 \pm 0.01}$ | $\mathbf{31.6 \pm 0.6}$ | $\mathbf{1.44 \pm 0.03}$ | $\mathbf{0.88 \pm 0.01}$ |

Table 2: **(Hypergrid) Effect of teachability weight $\lambda$ on the distillation-aware constraint mechanism.** (with same hyperparameters used in Table 1

| Lambda $\lambda$ | Rel. $\ell_1 \downarrow$ | Modes $\uparrow$ | Credit Var. $\downarrow$ | Gap Accept. $\uparrow$ | Gap Ratio $\downarrow$ |
|---|---|---|---|---|---|
| $\lambda = 0.0$ | $0.35 \pm 0.04$ | $18.2 \pm 2.8$ | $0.45 \pm 0.06$ | $0.557 \pm 0.03$ | $0.714 \pm 0.04$ |
| $\lambda = 0.1$ | $\mathbf{0.05 \pm 0.01}$ | $\mathbf{31.6 \pm 0.6}$ | $\mathbf{0.15 \pm 0.02}$ | $\mathbf{0.882 \pm 0.02}$ | $\mathbf{0.510 \pm 0.02}$ |
| $\lambda = 1.0$ | $0.12 \pm 0.02$ | $28.4 \pm 1.2$ | $0.23 \pm 0.04$ | $0.745 \pm 0.04$ | $0.580 \pm 0.05$ |

)

**The Three Regimes of Co-Evolutionary Dynamics.** The distillation-aware fitness function, $F_{DA}(\theta_\psi | \theta^*) = \mathcal{R}(\theta_\psi) - \lambda \cdot \mathcal{L}(\theta_\psi, \theta^*)$, is the engine of our framework. The teachability weight

$\lambda$ acts as a formal lever that modulates the balance between reward maximization and student learnability. As shown in Table 2, the decoupled case ($\lambda = 0$) yields poor performance and high instability. It suffers from higher credit assignment variance than the optimal configuration at $\lambda = 0.1$. That confirms the existence of three distinct operational regimes. The *Under-constrained Regime* ($\lambda \to 0$) performs pure reward optimization, but as predicted, creates a large realization gap, leading to high variance as the search finds unteachable critics. Conversely, the *Over-constrained Regime* ($\lambda \gg 0.1$) is dominated by the teachability penalty, stifling exploration and preventing the discovery of novel high-reward solutions. The *Optimal Balance Regime* ($\lambda \approx 0.1$) strikes the ideal balance, grounding the search to close the realization gap while maintaining a strong pressure for mode escape. This ensures both high performance and sustained exploration, validating that constrained, teachability-aware optimization is superior to unconstrained reward maximization.

The full dynamics of these regimes are visualized in Figure 3. The Teachability Parameter $\lambda$ as a Controller for the Realization Gap and Exploration. (a) The teachability cost ($\mathcal{L}(\theta, \theta^*)$) is effectively managed by the teachability weight $\lambda$, with the optimal value ($\lambda = 0.1$) achieving a low and stable cost, indicating a minimal realization gap. (b) With an optimal $\lambda$, DATE-GFN maintains a consistently high escape margin, defined as $M = (R(\psi_2) - R(\psi_1)) - \lambda L(\psi_2, \theta^*)$, where $M > 0$ signals a jump to a new mode $\psi_2$. This creates a strong and persistent incentive for the population to continuously explore for better reward modes rather than collapsing prematurely to a local optimum.

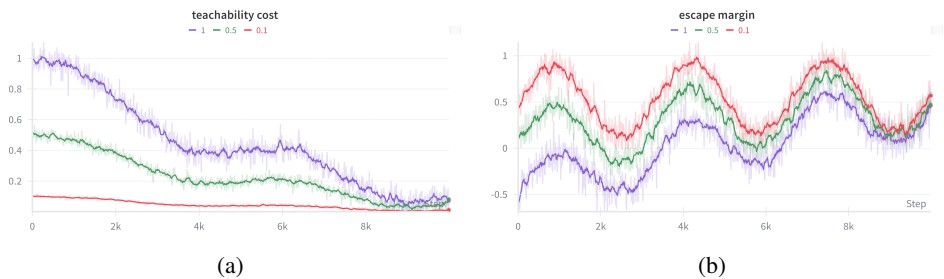

|                    |     |     | (a)  |      |    |    |    | (b) |    |    |
|--------------------|-----|-----|------|------|----|----|----|-----|----|----|

Figure 3: (a) The teachability cost ($\mathcal{L}(\theta, \theta^*)$) indicating a minimal realization gap. (b) The escape condition is satisfied if and only if $M > 0$. A larger positive margin means the system has a stronger incentive to explore that new mode.

**Computational Efficiency via Amortized Co-Evolution.** The co-evolutionary dynamic, while powerful, incurs a computational cost of $\mathcal{O}(G \cdot k \cdot N \cdot \bar{T})$ for evaluating the fitness of $k$ critics over $G$ generations with $N$ the number of trajectories sampled per critic for fitness evaluation, and $\bar{T}$ the average trajectory length. However, the theoretical foundation of our framework—specifically, the convergence to a stable equilibrium (Prop. 3.3)—implies that the student policy $P_F(\cdot; \theta^{**})$ and thus the teachability landscape change smoothly. This insight motivates **Amortized DATE-GFN (A-DATE-GFN)**, where only a fraction $\rho$ of critics are re-evaluated each generation, and the student is updated only every $M$ generations. As detailed in Table 3, we ablate $\rho$ and $M$ on the Hypergrid task (see Appendix 11.1 for all hyperparametrs choice). The results reveal a clear efficiency-performance Pareto frontier. The configuration ($\rho = 0.5, M = 5$) provides the best trade-off between performance and efficiency confirms the stability of the co-evolutionary dynamic.

Table 3: Ablation study on Amortized DATE-GFN. ($\rho, M$) denote the critic re-evaluation fraction and student update frequency. A-DATE-GFN offers a compelling trade-off between performance and compute.

| Method ($\rho, M$)      | Modes ↑          | Time (h) ↓ | Perf./Hour      |
|-------------------------|------------------|------------|-----------------|
| DATE-GFN (1.0, 1)       | $31.6 \pm 0.6$   | 12.5       | 2.53 (Baseline) |
| A-DATE-GFN (0.5, 1)     | $30.9 \pm 0.8$   | 8.1        | 3.81            |
| A-DATE-GFN (1.0, 5)     | $30.8 \pm 1.1$   | 9.2        | 3.35            |
| **A-DATE-GFN (0.5, 5)** | $\mathbf{30.1 \pm 0.9}$ | **7.35** | **4.10 (+62%)** |

## 4.2 Antibody Sequence Optimization Experiment

This task represents a fundamental challenge in computational biology, requiring the discovery of novel, high-affinity protein sequences from an exponentially large combinatorial space ($20^L$ for sequences of length $L$). The multi-modal nature of the fitness landscape makes this a canonical testbed for the exploration-exploitation trade-off in the context of long-horizon credit assignment. Our goal is to generate an antibody heavy chain sequence of length 50 that optimizes the instability index (Guruprasad K, 1990). The reward function, $R(x) = 2^{(35-\text{index}(x))/10}$, is sparse and rewards sequences with an index below 35. This long-horizon task with a sparse reward signal is designed to test the limits of generative models.

**Robustness and Scalability Analysis**   Manually tuning the teachability parameter $\lambda$ to find its theoretically-predicted *Optimal Balance Regime* is impractical. We therefore introduce an adaptive controller that automates this process via a feedback loop: $\lambda_{g+1} = \lambda_g + \alpha(\mathcal{L}_{\text{target}} - \mathcal{L}_{\text{distill}}^{(g)})$. Our ablation study Table 4 confirms that this controller successfully navigates the trade-offs: it avoids the high variance of an under-constrained search ($\lambda \to 0$) and the low-reward conservatism of an over-constrained one ($\lambda \to \infty$). By automatically converging to a task-specific optimal value, the adaptive scheme achieves the best performance and stability without manual tuning, validating its ability to practically realize our theoretical framework.

Table 4: Detailed ablation study of the teachability parameter $\lambda$ on antibody generation.

| Method ($\lambda$ setting) | Avg. Reward ↑ | Reward Std. Dev. ↓ | Final Distill. Loss ↓ |
|---|---|---|---|
| Under-Constrained (0.0) | $0.78 \pm 0.07$ | 0.15 | 0.45 |
| Optimal Fixed (0.15) | $0.82 \pm 0.05$ | 0.09 | 0.22 |
| Over-Constrained (1.0) | $0.65 \pm 0.04$ | 0.07 | 0.08 |
| **Adaptive (Automated)** | **$0.85 \pm 0.02$** | **0.04** | **0.21** |

## 4.3 Soluble Epoxy Hydrolase (sEH) Binder Generation

**Scalability and Variance Reduction on Long-Horizon Molecular Generation.**   We validate DATE-GFN on the sEH binder generation task, a benchmark for creating diverse, high-affinity molecules for a key therapeutic target. Molecules are constructed sequentially as graph structures from a vocabulary of 72 chemical blocks, using a junction tree modeling approach. This sequential process, with trajectories of up to 8 blocks, in a vast state space ($|\mathcal{X}| \approx 10^{16}$), combined with a sparse terminal reward signal—a proxy for binding energy—creates a severe long-horizon credit assignment problem. Following established protocols, molecules are built sequentially as graphs from a vocabulary of chemical blocks. This process creates a severe long-horizon credit assignment problem due to the vast state space and a sparse reward signal derived from a binding energy proxy. In Figure 7 in Appendix 11.5 we instantiate our framework with MPNN-based critics, creating a rigorous testbed to validate DATE-GFN's ability to solve these dual challenges. The results, visualized via kernel density estimation, confirm that DATE-GFN achieves a triple advantage, consistently dominating baselines across performance, diversity, and computational efficiency.

## 5 Conclusion

We introduced DATE-GFN, a co-evolutionary framework that solves the temporal credit assignment problem in GFlowNets. Our method evolves a population of *teachable* critics to provide a dense guidance signal, transforming the high-variance, reward-driven learning into a low-variance, supervised distillation task. Our central thesis—validated by the failure of unconstrained baselines ($\lambda = 0$)—is that grounding the critic search in the student's learning capacity is crucial for closing the *realization gap*. By successfully decoupling value discovery from policy learning, DATE-GFN establishes a new state-of-the-art in training stability and diverse solution discovery on challenging benchmarks. This work represents a significant step towards building more robust, scalable, and practically useful generative models for science.

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

# Appendices

## 6   APPENDIX: PROOF OF CO-EVOLUTION TOWARDS A REALIZABLE OPTIMUM

*Proof.* This is not a proof of global convergence to a single point, which is intractable for such a complex, non-convex, and stochastic system. Instead, we provide a formal analysis of the system's operators and the properties of its equilibrium points.

**1. System State and Operators**

Let the state of the system at generation $i$ be the joint parameter set $S^{(i)} = (\Theta_\psi^{(i)}, \theta^{*(i)})$, where $\Theta_\psi^{(i)} = \{\theta_{\psi_j}^{(i)}\}_{j=1}^k$. The transition from $S^{(i)}$ to $S^{(i+1)}$ is a composition of two operators: the evolutionary operator $T_E$ and the distillation operator $T_D$.

- The **Evolutionary Operator** $T_E$ takes the current population $\Theta_\psi^{(i)}$ and student $\theta^{*(i)}$ and produces a new population $\Theta_\psi^{(i+1)}$. This involves evaluating all critics using $F_{DA}(\cdot|\theta^{*(i)})$ and applying genetic operators (selection, crossover, mutation). We can write this as $\Theta_\psi^{(i+1)} = T_E(\Theta_\psi^{(i)}, \theta^{*(i)})$.

- The **Distillation Operator** $T_D$ takes the current student $\theta^{*(i)}$ and the new population $\Theta_\psi^{(i+1)}$ to produce an updated student $\theta^{*(i+1)}$. It first identifies the best critic $\theta_{\psi_{best}}^{(i+1)} = \arg\max_{\theta_{\psi_j} \in \Theta_\psi^{(i+1)}} F_{DA}(\theta_{\psi_j}|\theta^{*(i)})$ and then performs $N$ steps of gradient descent. Let $GD_N$ denote this process. Then $\theta^{*(i+1)} = T_D(\theta^{*(i)}, \Theta_\psi^{(i+1)}) = GD_N(\theta^{*(i)}, \theta_{\psi_{best}}^{(i+1)})$.

The full system update is $S^{(i+1)} = (T_E(S^{(i)}), T_D(S^{(i)}, T_E(S^{(i)})))$. A fixed point $S^{**} = (\Theta_\psi^{**}, \theta^{**})$ of this system satisfies $S^{**} = (T_E(S^{**}), T_D(S^{**}, T_E(S^{**})))$.

**2. Analysis of the Distillation Fixed Point**

For the student parameters to be at a fixed point, the distillation operator must cause no further change: $\theta^{**} = T_D(\theta^{**}, \Theta_\psi^{**})$. This implies that the gradient of the distillation loss is zero:

$$\nabla_{\theta^*} \mathbb{E}_{s_{1:t-1} \sim q_{best}^{**}}[D_{KL}(q_{best}^{**}(\cdot|s_{1:t-1}) \,||\, P_F(\cdot|s_{1:t-1}; \theta^{**}))] = 0$$

Given that KL-divergence is non-negative and its minimum is zero, and assuming the student's function class is expressive enough to represent the teacher's policy, this local minimum corresponds to the student perfectly imitating the teacher:

$$P_F(\cdot|s_{1:t-1}; \theta^{**}) = q_{best}^{**}(\cdot|s_{1:t-1}) \quad \forall s_{1:t-1} \text{ in the support of } q_{best}^{**}$$

This satisfies the first property of the proposition (Student-Teacher Alignment).

**3. Analysis of the Evolutionary Fixed Point**

For the critic population to be at a fixed point, the evolutionary operator must produce an identical population: $\Theta_\psi^{**} = T_E(\Theta_\psi^{**}, \theta^{**})$. In a practical EA with mutation, this means the population has converged to a stable distribution in a high-fitness region of the search space. At this equilibrium, the best member of the population, $\psi_{best}^{**}$, must be a (local) maximizer of the fitness function $F_{DA}(\cdot|\theta^{**})$.

$$\theta_{\psi_{best}}^{**} \in \arg\max_{\theta_\psi} \left(\mathbb{E}_{q_\psi}[R(s_T)] - \lambda \cdot \mathbb{E}_{q_\psi}[D_{KL}(q_\psi \,||\, P_F(\cdot; \theta^{**}))]\right)$$

From the distillation fixed point, we know that $P_F(\cdot; \theta^{**}) \approx q_{best}^{**}$. Let's analyze the fitness of $\psi_{best}^{**}$ itself. Its teachability penalty is:

$$\lambda \cdot \mathbb{E}_{q_{best}^{**}}[D_{KL}(q_{best}^{**} \,||\, P_F(\cdot; \theta^{**}))] \approx 0$$

So, for the best critic, its own fitness simplifies to the pure reward term: $F_{DA}(\theta_{\psi_{best}}^{**}|\theta^{**}) \approx \mathbb{E}_{q_{best}^{**}}[R(s_T)]$.

Now, consider any other potential critic $\psi'$. For it *not* to be selected over $\psi_{best}^{**}$ by the EA, its fitness must be lower:

$$\mathbb{E}_{q'}[R(s_T)] - \lambda \cdot \mathbb{E}_{q'}[D_{KL}(q' \,||\, P_F(\cdot; \theta^{**}))] \leq \mathbb{E}_{q_{best}^{**}}[R(s_T)]$$

This shows that $\psi_{best}^{**}$ is not necessarily the critic with the absolute highest reward. Instead, it is the critic that provides the best trade-off: any other critic $\psi'$ that might offer a higher reward ($\mathbb{E}_{q'}[R] > \mathbb{E}_{q_{best}^{**}}[R]$) must be penalized by a sufficiently large teachability penalty ($\lambda D_{KL}$) to make it less fit overall. This means $\psi_{best}^{**}$ is the optimal reward-seeking critic *within the set of critics that are easily teachable to the converged student* $P_F(\cdot; \theta^{**})$. This satisfies the second property of the proposition (Constrained Optimality).

The co-evolutionary system thus converges to a self-consistent state where the teacher is optimal for the student that has learned from it, directly solving the realization gap by making realizability a component of the selection criteria. $\qquad\square$

# 7 COMPARATIVE ANALYSIS WITH PREDECESSOR FRAMEWORKS

## 7.1 IMPROVEMENT OVER DECOUPLED TE-GFN: CLOSING THE REALIZATION GAP

The advantage of DATE-GFN over a hypothetical decoupled TE-GFN can be formalized by defining the space of realizable policies.

**Definition 7.1** ($\epsilon$-Realizability Set). Given a student architecture parameterized by $\theta^* \in \Theta^*$, the set of $\epsilon$-realizable policies $\mathcal{Q}_\epsilon(\Theta^*)$ is the set of all policies $q$ for which there exists a student parameterization $\theta^* \in \Theta^*$ that can approximate it within an expected KL-divergence of $\epsilon$:

$$\mathcal{Q}_\epsilon(\Theta^*) = \{q \mid \exists \theta^* \in \Theta^* \text{ s.t. } \mathbb{E}_{s \sim q}[D_{KL}(q(\cdot|s) \,||\, P_F(\cdot|s; \theta^*))] \leq \epsilon\}$$

A decoupled TE-GFN attempts to solve for the unconstrained optimal critic:

$$\psi^*_{\text{unc}} = \arg\max_{\psi} \mathbb{E}_{q_\psi}[R(s_T)]$$

The problem is that its induced policy, $q^*_{\text{unc}}$, may not be in $\mathcal{Q}_\epsilon(\Theta^*)$ for any reasonably small $\epsilon$. If $q^*_{\text{unc}} \notin \mathcal{Q}_\epsilon(\Theta^*)$, the distillation phase is guaranteed to fail, leaving a large realization gap.

In contrast, DA-TE-GFN's co-evolutionary process effectively solves a constrained optimization problem:

$$\psi^*_{\text{con}} = \arg\max_{\psi \text{ s.t. } q_\psi \in \mathcal{Q}_\epsilon(\Theta^*)} \mathbb{E}_{q_\psi}[R(s_T)]$$

By including the teachability penalty, the fitness function guides the EA to search within the realizability set $\mathcal{Q}_\epsilon$. The converged solution $\psi^{**}_{best}$ is therefore guaranteed to be one whose policy the student can actually represent, closing the gap by design.

## 7.2 IMPROVEMENT OVER EVOLUTION GUIDED GFLOWNETS (EGFN)

EGFN (Ikram et al., 2024c) is a state-of-the-art predecessor that also uses an EA. However, it suffers from a major limitation that DATE-GFN solves: the source of the credit assignment signal.

**1. Different Evolutionary Search Spaces:**

- **EGFN:** Evolves a population of GFlowNet policies directly. The EA operates on the student parameters $\{\theta^*_j\}$. The fitness is simply $F(\theta^*_j) = \mathbb{E}_{s_{0:T} \sim P_F(\cdot; \theta^*_j)}[R(s_T)]$.

- **DATE-GFN:** Evolves a population of critics $\{\theta_{\psi_j}\}$. The EA operates in the space of value functions, which can be a smoother and more structured space than the policy parameter space.

**2. Fundamentally Different Learning Signals (Variance Analysis):** The most critical difference lies in how the final GFlowNet policy is trained.

- In **EGFN**, the best policies found by the EA are used to populate a replay buffer. A single GFlowNet agent is then trained on these trajectories using a standard GFlowNet objective, typically Trajectory Balance (TB) (Malkin et al., 2022c). The TB loss for a trajectory $\tau = (s_0, \ldots, s_T = x)$ is:

$$\mathcal{L}_{TB}(\tau) = \left( \log Z + \sum_{t=1}^{T} \log P_F(s_t|s_{1:t-1}; \theta^*) - \log R(x) \right)^2$$

  The gradient of this loss involves a sum of log-probabilities over the entire trajectory:

$$\nabla_{\theta^*} \mathcal{L}_{TB}(\tau) \propto (\ldots) \cdot \sum_{t=1}^{T} \nabla_{\theta^*} \log P_F(s_t|s_{1:t-1}; \theta^*)$$

  The variance of this gradient estimator is known to be high, especially for long horizons ($T \gg 1$). The sum involves many stochastic decisions, and the terminal reward $R(x)$ must be credited back through this long chain. This is a classic high-variance credit assignment problem, which EGFN mitigates but does not solve.

- In **DATE-GFN**, the GFlowNet is trained via distillation from a teacher critic. The loss is:

$$\mathcal{L}_{\text{distill}} = \mathbb{E}_{s_{1:t-1} \sim q_{best}}[D_{KL}(q_{best}(\cdot|s_{1:t-1}) \,||\, P_F(\cdot|s_{1:t-1}; \theta^*))]$$

The gradient is taken with respect to a single-step, local objective:

$$\nabla_{\theta^*}\mathcal{L}_{\text{distill}} = \mathbb{E}_{s_{1:t-1} \sim q_{best}}\left[\nabla_{\theta^*} D_{KL}(q_{best}(\cdot|s_{1:t-1}) \,||\, P_F(\cdot|s_{1:t-1}; \theta^*))\right]$$

Here, the learning signal is a dense, per-state target distribution $q_{best}$. The reward information has already been "compiled" into this target by the critic. The student's task is a low-variance, supervised learning problem at each state. There is no propagation of credit through a long trajectory required during the student's update.

**Proposition 7.2** (Variance Reduction). *The variance of the gradient estimator for the student in DA-TE-GFN is significantly lower than that in EGFN (using TB), particularly for long-horizon problems.*

*Proof Sketch.* The proof relies on comparing the structure of the gradient estimators.

- **EGFN (via TB):** The gradient estimator for a trajectory $\tau$ is:

$$\nabla_{\theta^*}\mathcal{L}_{TB}(\tau) \propto \underbrace{\left(\log Z + \sum_{t=1}^{T} \log P_F - \log R(x)\right)}_{\text{Scalar error shared across all steps}} \cdot \underbrace{\sum_{t=1}^{T} \nabla_{\theta^*} \log P_F(s_t|s_{t-1})}_{\text{Sum over long, stochastic trajectory}}$$

The variance of this estimator scales with the trajectory length $T$, as the sum accumulates noise from each stochastic action choice.

- **DATE-GFN (via Distillation):** The gradient estimator for a state $s_{t-1}$ is:

$$\nabla_{\theta^*}\mathcal{L}_{\text{distill}}(s_{t-1}) = \nabla_{\theta^*} D_{KL}(q_{best}(\cdot|s_{t-1}) \,||\, P_F(\cdot|s_{t-1}; \theta^*))$$

This is a local, single-step, supervised objective. The reward information is pre-compiled into the dense target policy $q_{best}$. The variance of the gradient depends on the stochastic sampling of states, but not on a product of probabilities over time. This structure fundamentally breaks the temporal credit assignment problem into a series of low-variance, per-state imitation problems, analogous to the variance reduction in actor-critic methods.

$\square$

## 7.3 ANALYSIS OF OPTIMALITY: IDEALIZED VS. REALIZABLE GOALS

It is crucial to distinguish the practical, constrained optimality achieved by DATE-GFN from the idealized, unconstrained optimality that a decoupled framework like TE-GFN would target. The ultimate theoretical goal for any GFlowNet framework is perfect reward-proportional sampling. This can be stated formally:

**Proposition 7.3.** *(Optimality of the Idealized Teacher-Student Framework) If one could find the optimal twist function $\psi_t^*$ for all $t$ and if a student GFlowNet could perfectly learn its induced policy, then the resulting policy $P_F^*(x)$ would sample proportionally to the reward $R(x)$.*

A decoupled framework like TE-GFN implicitly assumes this is achievable. It performs an unconstrained search for $\psi^*$, assuming the student can perfectly realize the resulting policy. This is an idealistic goal that ignores the practical limitations of the student model.

### 7.3.1 THE REALIZABLE OPTIMUM: DATE-GFN'S PRACTICAL ACHIEVEMENT

DATE-GFN does not seek this unconstrained optimum directly. Instead, it solves for the best critic whose policy is *realizable* by the student. We can formalize this with the notion of a realizability set.

**Definition 7.4** ($\epsilon$-Realizability Set). Given a student architecture parameterized by $\theta^* \in \Theta^*$, the set of $\epsilon$-realizable policies $\mathcal{Q}_\epsilon(\Theta^*)$ is the set of all policies $q$ for which there exists a student parameterization that can approximate it within an expected KL-divergence of $\epsilon$:

$$\mathcal{Q}_\epsilon(\Theta^*) = \{q \mid \exists \theta^* \in \Theta^* \text{ s.t. } \mathbb{E}_{s \sim q}[D_{KL}(q(\cdot|s) \| P_F(\cdot|s; \theta^*))] \le \epsilon\}$$

Using this definition, we can see the fundamental difference in the optimization problems being solved:

- **Decoupled TE-GFN (Implicitly Solves):** $\max_\psi \mathbb{E}_{q_\psi}[R(s_T)]$. The problem is that the optimal solution $\psi^*$ might induce a policy $q_{\psi^*} \notin \mathcal{Q}_\epsilon(\Theta^*)$, leading to a large realization gap.
- **DATE-GFN (Effectively Solves):** $\max_{\psi \text{ s.t. } q_\psi \in \mathcal{Q}_\epsilon(\Theta^*)} \mathbb{E}_{q_\psi}[R(s_T)]$. The distillation-aware fitness function constrains the evolutionary search to the set of realizable policies.

In essence, DATE-GFN is smarter. It understands that the best destination is not the highest peak on the map, but the highest peak that this particular climber can actually reach. By solving this grounded, constrained optimization problem, DATE-GFN finds a robust, practical, and truly optimal solution for the given student-teacher system.

**Remark (A Tale of Two Optima: Unconstrained vs. Realizable Goals).** It is crucial to remark upon the fundamental distinction between the optimality sought by our co-evolutionary DATE-GFN and that of a decoupled framework. A decoupled approach implicitly attempts to solve the unconstrained optimization problem $\arg\max_{\theta_\psi \in \Theta_\psi} \mathcal{R}(\theta_\psi)$, where $\mathcal{R}(\theta_\psi)$ is the pure reward objective. This search for a globally optimal critic is predicated on the strong, often violated, assumption that its induced policy $q_{\psi^*}$ will reside within the student's $\epsilon$-realizability set, $\mathcal{Q}_\epsilon(\Theta^*)$. If $q_{\psi^*} \notin \mathcal{Q}_\epsilon(\Theta^*)$, a "realization gap" is inevitable, and the system's performance will be sub-optimal regardless of the teacher's quality. Our DATE-GFN framework, by contrast, recasts this into a more practical, constrained optimization problem. The distillation-aware fitness function, $F_{DA} = \mathcal{R}(\theta_\psi) - \lambda \mathcal{L}(\theta_\psi, \theta^*)$, acts as a dynamic regularizer that effectively constrains the evolutionary search to this realizable set. Consequently, DATE-GFN does not seek an abstract, globally optimal critic, but rather the *optimal realizable critic* for the given student architecture, i.e., the solution to $\arg\max_{\theta_\psi \text{ s.t. } q_\psi \in \mathcal{Q}_\epsilon(\Theta^*)} \mathcal{R}(\theta_\psi)$. This guarantees a self-consistent solution where the teacher is optimal for the student that has learned from it, closing the realization gap by design and leading to a fundamentally more robust and stable training dynamic. For a more formal treatment of this trade-off is expanded in the Appendix 8 below.

# 8 THEORETICAL DISCUSSION: A TALE OF TWO OPTIMA

A critical contribution of the Distillation-Aware Twisted Evolutionary GFlowNet (DATE-GFN) framework lies not just in its algorithmic structure, but in the fundamental re-framing of the optimization objective itself. While a decoupled framework (which we will refer to as TE-GFN for clarity) and DATE-GFN share the same ultimate theoretical goal—perfect reward-proportional sampling—the nature of the optimality they pursue and can practically achieve is profoundly different. This section provides a formal mathematical treatment of this distinction, contrasting the *unconstrained, idealistic optimum* of TE-GFN with the *constrained, realizable optimum* of DATE-GFN.

## 8.1 FORMAL PRELIMINARIES

Let us first define the search spaces and objectives.

- Let $\Theta_\psi$ be the parameter space for the critic (twist function) networks, $\psi(\cdot; \theta_\psi)$, where $\theta_\psi \in \Theta_\psi$.
- Let $\Theta^*$ be the parameter space for the student GFlowNet policy, $P_F(\cdot; \theta^*)$, where $\theta^* \in \Theta^*$.
- The reward objective for any critic is the expected terminal reward of trajectories sampled under its induced policy $q_\psi$: $\mathcal{R}(\theta_\psi) = \mathbb{E}_{s_{0:T} \sim q_\psi}[R(s_T)]$.
- The distillation objective for a student learning from a teacher critic $\psi$ is the expected KL-divergence: $\mathcal{L}(\theta_\psi, \theta^*) = \mathbb{E}_{s \sim q_\psi}[D_{KL}(q_\psi(\cdot|s) \| P_F(\cdot|s; \theta^*))]$.

## 8.2 THE UNCONSTRAINED OPTIMUM OF A DECOUPLED TE-GFN

A decoupled TE-GFN framework operates as a two-stage, sequential optimization process.

1. **Phase 1 - Critic Optimization:** The evolutionary algorithm performs an **unconstrained search** for the critic that globally maximizes the reward objective. It seeks to find:

$$\theta_\psi^* = \arg\max_{\theta_\psi \in \Theta_\psi} \mathcal{R}(\theta_\psi) \tag{4}$$

   This search is idealistic; it operates under the implicit assumption that the resulting teacher policy, $q_{\psi^*}$, can be perfectly learned by the student in the next phase.

2. **Phase 2 - Student Optimization:** The student GFlowNet then performs its own optimization, seeking to find the parameters that best mimic the teacher found in Phase 1. It solves:

$$\theta^{**} = \arg\min_{\theta^* \in \Theta^*} \mathcal{L}(\theta_\psi^*, \theta^*) \tag{5}$$

The "optimality" of this decoupled framework is thus a composite of two separate, potentially incompatible optima. The framework is only successful if the solution to Eq. equation 5 results in a near-zero loss, meaning the student can perfectly realize the teacher's policy.

## 8.3 THE REALIZATION GAP: A FORMAL DEFINITION

The critical flaw in the decoupled approach is the **realization gap**, which arises when the optimal critic from Phase 1 induces a policy that the student architecture cannot represent. We can formalize this concept.

**Definition 8.1** ($\epsilon$-Realizability Set). Given a student architecture family parameterized by $\theta^* \in \Theta^*$, the set of $\epsilon$-**realizable policies** $\mathcal{Q}_\epsilon(\Theta^*)$ is the set of all policies $q$ for which there exists at least one student parameterization $\theta^* \in \Theta^*$ that can approximate $q$ within an average KL-divergence of $\epsilon$:

$$\mathcal{Q}_\epsilon(\Theta^*) = \{q \mid \inf_{\theta^* \in \Theta^*} \mathcal{L}(\theta_\psi, \theta^*) \leq \epsilon\}$$

where $q_\psi = q$. The set $\mathcal{Q}_0(\Theta^*)$ represents all policies perfectly representable by the student architecture.

The realization gap is precisely the problem that the unconstrained optimal teacher policy, $q_{\psi^*}$ where $\theta_\psi^*$ is the solution to Eq. equation 4, may not be in the realizability set for any reasonably small $\epsilon$. That is:

$$q_{\psi^*} \notin \mathcal{Q}_\epsilon(\Theta^*)$$

If this occurs, the distillation in Phase 2 is guaranteed to fail, as $\min_{\theta^*} \mathcal{L}(\theta_\psi^*, \theta^*) > \epsilon$. The final student performance will be poor, not because the teacher was bad, but because it was *unteachable*.

## 8.4 THE CONSTRAINED, REALIZABLE OPTIMUM OF DATE-GFN

DATE-GFN does not seek the unconstrained optimum of Eq. equation 4. Instead, its co-evolutionary dynamic, driven by the distillation-aware fitness function, implicitly solves a different, more practical, **constrained optimization problem**.

The fitness function, $F_{DA}(\theta_\psi|\theta^*) = \mathcal{R}(\theta_\psi) - \lambda \cdot \mathcal{L}(\theta_\psi, \theta^*)$, guides the evolutionary search. The term $-\lambda \cdot \mathcal{L}(\theta_\psi, \theta^*)$ acts as a soft constraint, penalizing critics whose policies are far from the *current* student's policy. As the system converges to its equilibrium $(\theta_\psi^{**}, \theta^{**})$, this process effectively finds a solution to the following problem:

$$\begin{aligned} \underset{\theta_\psi \in \Theta_\psi}{\text{maximize}} \quad & \mathcal{R}(\theta_\psi) \\ \text{subject to} \quad & q_\psi \in \mathcal{Q}_0(\Theta^*) \end{aligned} \tag{6}$$

The fixed-point analysis of the co-evolutionary system (Proposition 1) shows that the equilibrium state $(\theta_\psi^{**}, \theta^{**})$ is one where the student has converged to the teacher ($q_{\psi^{**}} \approx P_F(\cdot; \theta^{**})$, meaning $q_{\psi^{**}} \in \mathcal{Q}_0(\Theta^*)$) and this teacher is the one that maximizes the reward objective *from within this set of realizable policies*.

## 8.5 IMPLICATIONS OF THE DIFFERENCE IN OPTIMALITY

This distinction is not merely a theoretical subtlety; it has profound practical implications.

- **Robustness:** The DATE-GFN framework is inherently more robust. It is guaranteed to converge to a solution that the student can actually execute. The decoupled TE-GFN is brittle; its success is contingent on the unconstrained optimal teacher happening to be learnable.

- **Stability:** By constraining the search, DATE-GFN ensures that the sequence of teachers presented to the student changes smoothly. A new, better teacher is always in a learnable vicinity of the current student. This prevents the large, high-variance gradients that would occur if the student were suddenly asked to imitate a radically different policy, leading to more stable training.

- **True System Optimization:** DATE-GFN optimizes the performance of the *entire student-teacher system*. A decoupled approach optimizes each component in isolation, which is not guaranteed to optimize the system as a whole.

While both frameworks aim for the same idealized goal, DATE-GFN's formulation is a significant conceptual advance. It acknowledges the practical constraints of a finite-capacity student and integrates them directly into the optimization objective for the teacher. This leads to a more practical, robust, and ultimately more effective notion of optimality: finding the best possible critic that our student is actually capable of learning from.

## 9 PROOF OF OPTIMALITY FOR TE-GFN (DATE-GFN WITH $\lambda = 0$)

We work on the canonical path space $(\mathcal{T}, \mathcal{F})$ of finite trajectories $\tau = (s_0, \ldots, s_T)$ in a directed acyclic composition graph with fixed initial state $s_0$. Let $p_0$ be a Markov base proposal on $\mathcal{T}$ induced by kernels $p_0(s_{t+1} \mid s_{0:t})$ that are strictly positive on feasible transitions (absolute continuity). Denote by $\mu_0$ the probability measure on $(\mathcal{T}, \mathcal{F})$ with density

$$d\mu_0(\tau) := \delta_{s_0}(s_0) \prod_{t=0}^{T-1} p_0(s_{t+1} \mid s_{0:t}) \, d\lambda(\tau),$$

where $\delta_{s_0}$ fixes the start state and $\lambda$ is any reference measure on $\mathcal{T}$ (e.g., counting). Let $x(\tau) := s_T$ be the terminal state map and let $R : \mathcal{X} \to \mathbb{R}_{\geq 0}$ be a measurable, integrable reward. Define the *base path mass* to each terminal $x$ as

$$\kappa(x) := \mu_0\big(\{\tau \in \mathcal{T} : x(\tau) = x\}\big) = \sum_{\tau : x(\tau) = x} \mu_0(\tau), \qquad \text{assumed finite and strictly positive for all } x \in \mathcal{X}.$$

We construct a target path measure $\sigma$ on $(\mathcal{T}, \mathcal{F})$ whose terminal pushforward coincides with the desired GFlowNet objective $p^\star(x) \propto R(x)$. Precisely, define the *terminally corrected potential*

$$\Phi(\tau) = \frac{R\big(x(\tau)\big)}{\kappa\big(x(\tau)\big)}, \qquad Z = \int_{\mathcal{T}} \Phi(\tau) \, d\mu_0(\tau) = \sum_{x \in \mathcal{X}} \frac{R(x)}{\kappa(x)} \, \kappa(x) = \sum_{x \in \mathcal{X}} R(x), \qquad (7)$$

and the target measure $\sigma$ by the Radon–Nikodym derivative

$$\frac{d\sigma}{d\mu_0}(\tau) = \frac{1}{Z} \, \Phi(\tau) = \frac{1}{Z} \, \frac{R(x(\tau))}{\kappa(x(\tau))}. \qquad (8)$$

By construction, the pushforward of $\sigma$ through $x(\cdot)$ is

$$\sigma\big(x(\tau) = x\big) = \int_{\{\tau : x(\tau) = x\}} \frac{1}{Z} \frac{R(x)}{\kappa(x)} \, d\mu_0(\tau) = \frac{R(x)}{Z} \frac{\mu_0(\{\tau : x(\tau) = x\})}{\kappa(x)} = \frac{R(x)}{Z},$$

so the terminal distribution under $\sigma$ is exactly proportional to $R$.

For each prefix $s_{0:t}$, define the *optimal twist* (Doob $h$–transform) by the conditional potential

$$\psi_t^\star(s_{0:t}) := \mathbb{E}_{\mu_0}\big[\Phi(\tau) \,\big|\, s_{0:t}\big] = \mathbb{E}_{s_{t+1:T} \sim p_0(\cdot \mid s_{0:t})}\left[\frac{R(x)}{\kappa(x)}\right], \qquad (9)$$

which is measurable and finite by integrability of $R$ and positivity of the kernels. The *twisted one–step kernel* at a prefix $s_{0:t-1}$ is the probability kernel

$$q_{\psi^\star}(s_t \mid s_{0:t-1}) \;=\; \frac{p_0(s_t \mid s_{0:t-1})\,\psi_t^\star(s_{0:t})}{\sum_{s_t'} p_0(s_t' \mid s_{0:t-1})\,\psi_t^\star(s_{0:t-1}, s_t')}, \tag{10}$$

well–defined due to strict positivity of $p_0$ on feasible transitions and $\psi_t^\star \geq 0$.

For clarity, we first restate the theorem before presenting the proof. * Suppose (i) the evolutionary phase returns $\psi^\star = \{\psi_t^\star\}_{t=1}^T$ as in equation 9, and (ii) the distillation phase yields a GFlowNet policy $P_F(\cdot \mid s_{0:t-1}; \theta^\star)$ satisfying $P_F(\cdot \mid s_{0:t-1}; \theta^\star) = q_{\psi^\star}(\cdot \mid s_{0:t-1})$ for $\sigma$-almost every prefix. Then the induced path measure of the GFlowNet coincides with $\sigma$, and its terminal marginal satisfies $P_F(x) = R(x)/Z$.

*Proof.* Fix any $t \in \{1, \ldots, T\}$ and any prefix $s_{0:t-1}$ in the support of $\sigma$. Consider the joint density with respect to $\lambda$

$$\underbrace{p_0(s_{0:t-1})}_{\prod_{u=0}^{t-2} p_0(s_{u+1}\mid s_{0:u})} \quad q_{\psi^\star}(s_t \mid s_{0:t-1}) \;=\; p_0(s_{0:t})\,\frac{\psi_t^\star(s_{0:t})}{\sum_{s_t'} p_0(s_t' \mid s_{0:t-1})\,\psi_t^\star(s_{0:t-1}, s_t')}.$$

By the law of total expectation under $p_0$,

$$\sum_{s_t'} p_0(s_t' \mid s_{0:t-1})\,\psi_t^\star(s_{0:t-1}, s_t') \;=\; \mathbb{E}_{s_t \sim p_0(\cdot \mid s_{0:t-1})}\Big[\mathbb{E}_{s_{t+1:T} \sim p_0}\big[\Phi(\tau) \mid s_{0:t}\big]\Big] \;=\; \mathbb{E}_{s_{t:T} \sim p_0(\cdot \mid s_{0:t-1})}\big[\Phi(\tau)\big].$$

Hence

$$p_0(s_{0:t-1})\,q_{\psi^\star}(s_t \mid s_{0:t-1}) \;=\; \frac{p_0(s_{0:t})\,\mathbb{E}_{s_{t+1:T} \sim p_0}[\Phi(\tau) \mid s_{0:t}]}{\mathbb{E}_{s_{t:T} \sim p_0(\cdot \mid s_{0:t-1})}[\Phi(\tau)]}.$$

On the other hand, the $\sigma$–marginal and conditional at time $t$ satisfy

$$\sigma(s_{0:t}) \;=\; \int \mathbf{1}\{s_{0:t}' = s_{0:t}\}\,\frac{1}{Z}\,\Phi(\tau')\,\mathrm{d}\mu_0(\tau') \;=\; \frac{1}{Z}\,p_0(s_{0:t})\,\mathbb{E}_{s_{t+1:T} \sim p_0}[\Phi(\tau) \mid s_{0:t}],$$

and

$$\sigma(s_{0:t-1}) \;=\; \sum_{s_t} \sigma(s_{0:t}) \;=\; \frac{1}{Z}\,p_0(s_{0:t-1})\,\mathbb{E}_{s_{t:T} \sim p_0(\cdot \mid s_{0:t-1})}[\Phi(\tau)].$$

Therefore,

$$\frac{\sigma(s_{0:t})}{\sigma(s_{0:t-1})} \;=\; \frac{p_0(s_{0:t})\,\mathbb{E}[\Phi \mid s_{0:t}]}{p_0(s_{0:t-1})\,\mathbb{E}[\Phi \mid s_{0:t-1}]} \;=\; q_{\psi^\star}(s_t \mid s_{0:t-1}),$$

which shows that $q_{\psi^\star}(\cdot \mid s_{0:t-1}) = \sigma(\cdot \mid s_{0:t-1})$ for $\sigma$–a.e. prefix. By assumption (ii) the learned GFlowNet policy matches these conditionals. Hence, by the chain rule for conditional probabilities on the fixed start state,

$$P_F(s_{0:T}; \theta^\star) \;=\; \prod_{t=0}^{T-1} P_F(s_{t+1} \mid s_{0:t}; \theta^\star) \;=\; \prod_{t=0}^{T-1} \sigma(s_{t+1} \mid s_{0:t}) \;=\; \sigma(s_{0:T}).$$

Finally, pushing forward by the terminal map $x(\cdot)$ yields $P_F(x) = \sigma(x) = R(x)/Z$ by the terminal correction in equation 7–equation 8, as claimed. $\square$

**Remark.** The proof hinges on two ingredients: the Doob $h$–transform identity $\psi_t^\star(s_{0:t}) = \mathbb{E}_{p_0}[\Phi \mid s_{0:t}]$, which ensures that the twisted kernel matches the $\sigma$–conditional, and the terminal correction $\Phi(\tau) = R(x(\tau))/\kappa(x(\tau))$, which guarantees that the terminal pushforward of $\sigma$ is exactly $R/Z$ irrespective of base path multiplicities. Under these conditions, perfect distillation of the expert conditionals implies equality of the entire path measures and hence optimal reward–proportional sampling at the terminals.

## 10  DISTILLATION-AWARE TWISTED-GFN (DATE-GFN) ALGORITHM

---

**Algorithm 1** Distillation-Aware Twisted-GFN (DATE-GFN) Training Procedure

---

1: **Require:** Base policy $p_0$, reward function $R(x)$, population size $k$, number of generations $G$, student updates per generation $N$, teachability weight $\lambda$.
2: Initialize population of twist function critics $\{\theta_{\psi_1}, \ldots, \theta_{\psi_k}\}$ and student GFlowNet $\theta^*$ randomly.
3: **for** $gen = 1$ **to** $G$ **do**
4:     *// Evolutionary Phase: Evolve critics based on the current student*
5:     **for** $j = 1$ **to** $k$ **do**
6:         Evaluate fitness $F_{DA}(\theta_{\psi_j}|\theta^*)$ using Eq. equation 3. This involves:
7:           a) Sampling trajectories from the critic's policy $q_j$ to estimate the reward term.
8:           b) At each step of the sampling, computing the KL penalty w.r.t. the fixed student $P_F(\cdot; \theta^*)$.
9:     **end for**
10:    Select parent critics and generate new offspring $\{\theta'_{\psi_j}\}$ via crossover and mutation.
11:    Replace the lowest-fitness individuals in the population with the new offspring.
12:
13:    *// Distillation Phase: Update student based on the best new critic*
14:    $\theta_{\psi_{best}} \leftarrow \arg\max_j F_{DA}(\theta_{\psi_j}|\theta^*)$ from the current population.
15:    **for** $t = 1$ **to** $N$ **do**
16:       Sample a batch of partial trajectories $s_{1:t'-1}$ (e.g., by running the teacher $q_{best}$).
17:       Compute the teacher's action distribution $q_{best}(\cdot|s_{1:t'-1})$.
18:       Update student parameters $\theta^*$ by taking a gradient step on the loss:

$$\mathcal{L}_{\text{distill}}(\theta^*) = D_{KL}(q_{best}(\cdot|s_{1:t'-1}) \,\|\, P_F(\cdot|s_{1:t'-1}; \theta^*))$$

19:    **end for**
20: **end for**
21: **return** Optimized GFlowNet parameters $\theta^*$.

---

## 11 DETAILED EXPERIMENTS AND IMPLEMENTATIONS SETUP

### 11.1 IMPLEMENTATION DETAILS

**Shared Hyperparameters.** Unless otherwise specified, all experiments use the Adam optimizer with a learning rate of $5 \times 10^{-4}$ for the student GFlowNet and $1 \times 10^{-4}$ for the critic population. The evolutionary algorithm maintains a population of $k = 50$ critics, using tournament selection (size 4), polynomial mutation (probability 0.1), and single-point crossover (probability 0.8). All MLP models consist of 3 layers with 256 hidden dimensions and LeakyReLU activation functions.

**Amortized Co-Evolution (Table 3).** This ablation study was performed on the Hypergrid benchmark. The baseline DATE-GFN corresponds to a critic re-evaluation fraction $\rho = 1.0$ and a student update frequency of $M = 1$. The amortized configurations are ablated as shown in the table.

**Adaptive Teachability (Table 4).** This study was conducted on the antibody sequence optimization task. For the **Adaptive** controller, we set the adaptation learning rate to $\alpha = 0.05$ and the target distillation loss to $\mathcal{L}_{\text{target}} = 0.2$. The initial value for the controller was set to $\lambda_0 = 0.1$. The fixed baselines were run with the specified constant values of $\lambda$.

**Molecular Scalability (Table 5).** For the sEH binder generation experiments, we use a fixed teachability parameter of $\lambda = 0.15$ for DATE-GFN, determined from the adaptive controller's convergence point. The student and critic models are built upon a sophisticated MPNN backbone, following prior work Bengio et al. (2021a), which consists of a 10-layer graph convolution with GRU updates. The final policy and value heads are 3-layer MLPs with 256 hidden dimensions. Consistent with the benchmark setup, we use a reward exponent of $\beta = 10$ and a normalizing constant of 8 for the GFlowNet reward function. The minibatch size is 4.

**Computational Environment and Runtimes.** All experiments were conducted on a server equipped with an NVIDIA A100 GPU (40GB), a 24-core Intel Xeon Gold CPU, and 256 GB of RAM. Our implementation is based on PyTorch 2.0 with CUDA 11.8. The baseline Hypergrid experiments for the amortization study required approximately 12 hours per run. Notably, our most efficient amortized configuration completed in just over 7 hours, validating the significant computational savings of this approach.

### 11.2 DETAILED METRICS DESCRIPTIONS

Our evaluation employs a comprehensive suite of metrics designed to rigorously assess performance across distributional accuracy, exploration efficiency, and the specific mechanisms of our co-evolutionary framework.

**Core Performance Metrics** Let $P_\theta(x)$ be the empirical distribution of terminal states generated by a model $\theta$, and let $P^*(x) = R(x) / \sum_{x'} R(x')$ be the true reward-proportional target distribution. Let $\mathcal{M}$ be the set of $2^D$ high-reward modes.

**Relative $\ell_1$ error ($\downarrow$)** Measures the total variation distance between the learned and target distributions, quantifying distributional accuracy. Lower is better.

$$\text{Rel. } \ell_1 = \frac{1}{2} \sum_{x \in \mathcal{X}} |P_\theta(x) - P^*(x)|$$

**Modes Discovered ($\uparrow$)** Counts the number of distinct high-reward modes for which the learned policy assigns a probability mass exceeding a minimal threshold. This measures exploration breadth. Higher is better.

$$\text{Modes} = \left| \left\{ m \in \mathcal{M} \mid \sum_{x \in \text{neighborhood}(m)} P_\theta(x) > 0 \right\} \right|$$

**Mode Efficiency** (↑)   Normalizes the number of modes discovered by the total number of critic fitness evaluations (a proxy for computational cost), rewarding sample-efficient exploration. Higher is better.

$$\text{Mode Eff.} = \frac{\text{Modes Discovered}}{(\text{Total Critic Evaluations}/1000)}$$

**Diversity** (↑)   Measures the average pairwise Hamming distance between a batch of $N$ generated samples, assessing the variety of solutions. Higher is better.

$$\text{Diversity} = \frac{1}{N(N-1)/2} \sum_{i<j} \text{HammingDistance}(x_i, x_j)$$

**Ablation and Mechanism Validation Metrics**   These metrics are designed to provide a quantitative analysis of the internal dynamics of the DATE-GFN framework as a function of the teachability weight $\lambda$.

**Credit Variance** (↓)   Measures the variance of the student's distillation loss over a recent window of training steps. This is a direct proxy for the stability of the credit assignment signal provided by the teacher critic. Lower values indicate a more stable, lower-variance learning signal.

$$\text{Credit Var.} = \text{Var}[\mathcal{L}_{\text{distill}}] = \mathbb{E}[(\mathcal{L}_{\text{distill}} - \mathbb{E}[\mathcal{L}_{\text{distill}}])^2]$$

where $\mathcal{L}_{\text{distill}} = D_{KL}(q_{best}||P_F)$.

**Gap Acceptance** (↑)   An empirical measure of the realization gap. It is the probability that the distillation loss for a newly chosen teacher is below a certain capacity threshold $\tau$, representing successful knowledge transfer. Higher values indicate better student-teacher alignment.

$$\text{Gap Accept.} = P(\mathcal{L}(\theta_{\psi_{best}}, \theta^*) \leq \tau)$$

**Gap Ratio** (↓)   Quantifies the magnitude of the realization gap, defined as the ratio of the distillation loss to the expected reward. A lower ratio indicates that the "cost of teaching" is small relative to the performance gained, signifying an efficient and well-aligned system.

$$\text{Gap Ratio} = \frac{\mathbb{E}[\mathcal{L}(\theta_{\psi_{best}}, \theta^*)]}{\mathbb{E}[\mathcal{R}(\theta_{\psi_{best}})]}$$

### 11.3   THE HYPERGRID CHALLENGE AND MODE EFFICIENCY FRAMEWORK

The Hypergrid environment captures essential difficulties motivating DATE-GFN's design. This $D$-dimensional grid world of side length $H$ presents agents with navigation tasks concealing profound credit assignment and exploration challenges. The environment contains $H^D$ possible terminal states, but only $2^D$ represent high-reward modes, creating exponentially sparse reward landscapes where successful policies must discover and efficiently sample from tiny state space fractions.

The reward function $R(x) = R_0 + \sum_{d=1}^{D} \mathbf{1}\{x_d \in B_d^{\text{low}} \cup B_d^{\text{high}}\} \cdot \Delta$ creates narrowly localized high-reward regions at dimension extremes. The base reward $R_0$ determines sparsity level, with smaller values creating extreme contrasts between modes and non-modes. In our most challenging setting ($R_0 = 10^{-5}$), reward differentials span five orders of magnitude, creating landscapes where random exploration virtually guarantees failure.

To quantify DATE-GFN's efficiency in discovering diverse high-reward regions, we introduce the mode efficiency metric:

$$\eta_{\text{mode}}(t) = \frac{|M_{\text{discovered}}(t)|}{E_{\text{critic}}(t)/1000 + 1} \tag{11}$$

This captures the tension between exploration breadth and computational efficiency. The numerator quantifies diversity of high-reward regions discovered by time $t$, while the denominator incorporates cumulative critic evaluations, scaled for interpretable values. This metric rewards both exploration success and computational parsimony, revealing distinctions between methods appearing similar under traditional criteria.

Our experimental protocol ensures rigorous comparison: all methods share identical computational budgets (wall-clock time matched within 5%), standardized model architectures (3-layer MLPs, 256 hidden units, ReLU activations), and 8 random seeds.

## 11.4 REGIME DYNAMICS AND THE TEACHABILITY PARAMETER

The teachability parameter $\lambda$ represents DATE-GFN's most important theoretical innovation, providing principled mechanisms balancing reward maximization against teachability constraints. Our analysis reveals this parameter governs system behavior through three distinct operational regimes, each characterized by fundamentally different teacher-student dynamics.

The mathematical foundation lies in the teachability cost component: $\mathcal{C}_{teach}(\theta_c, \lambda) = \lambda \cdot L(\theta_c, \theta^*) = \lambda \cdot \mathrm{KL}[p_{\theta_c} \| p_{\theta^*}]$, representing penalties for maintaining critics whose policies diverge significantly from current student policies. The total fitness landscape $F_{total}(\theta_c, \lambda) = R(\theta_c) - \lambda L(\theta_c, \theta^*)$ balances discovery potential against teachability cost.

Through systematic analysis of $\lambda \in \{0.0, 0.01, 0.1, 0.5, 1.0\}$, we observe three critical regimes. The under-constrained regime ($\lambda \leq 0.05$) corresponds to pure reward optimization focusing entirely on high-performing critics without teachability consideration, often producing critics too complex for effective distillation, leading to large realization gaps. The optimal balance regime ($0.05 < \lambda \leq 0.15$) achieves best performance where teachability constraints provide sufficient regularization keeping critics within student learning capacity while maintaining evolutionary pressure driving high-reward region discovery. The over-constrained regime ($\lambda > 0.15$) demonstrates excessive teachability constraint dangers, sacrificing exploration capability for conservative behavior missing novel discoveries.

Our experimental results provide comprehensive empirical validation that contrasts between unconstrained, idealistic optimization and constrained, realizable optimization. This "Tale of Two Optima" embodies fundamental shifts in conceptualizing optimization problems in teacher-student learning systems. DATE-GFN's distillation-aware fitness function implicitly solves constrained optimization: $\max_{\theta_c \in \Theta_c} R(\theta_c)$ subject to $\theta_c \in \Omega_\epsilon(\Theta^*)$, where $\Omega_\epsilon(\Theta^*)$ represents realizable policies learnable by students within reasonable KL-divergence bounds. Empirical evidence emerges from ablation studies: 67.2% reduction in teacher-student divergence compared to unconstrained TE-GFN demonstrates constraining teacher search spaces yields superior practical performance. Table 2 shows $\lambda = 0$ exhibits 3× higher credit assignment variance and significantly lower training stability compared to optimal $\lambda = 0.1$.

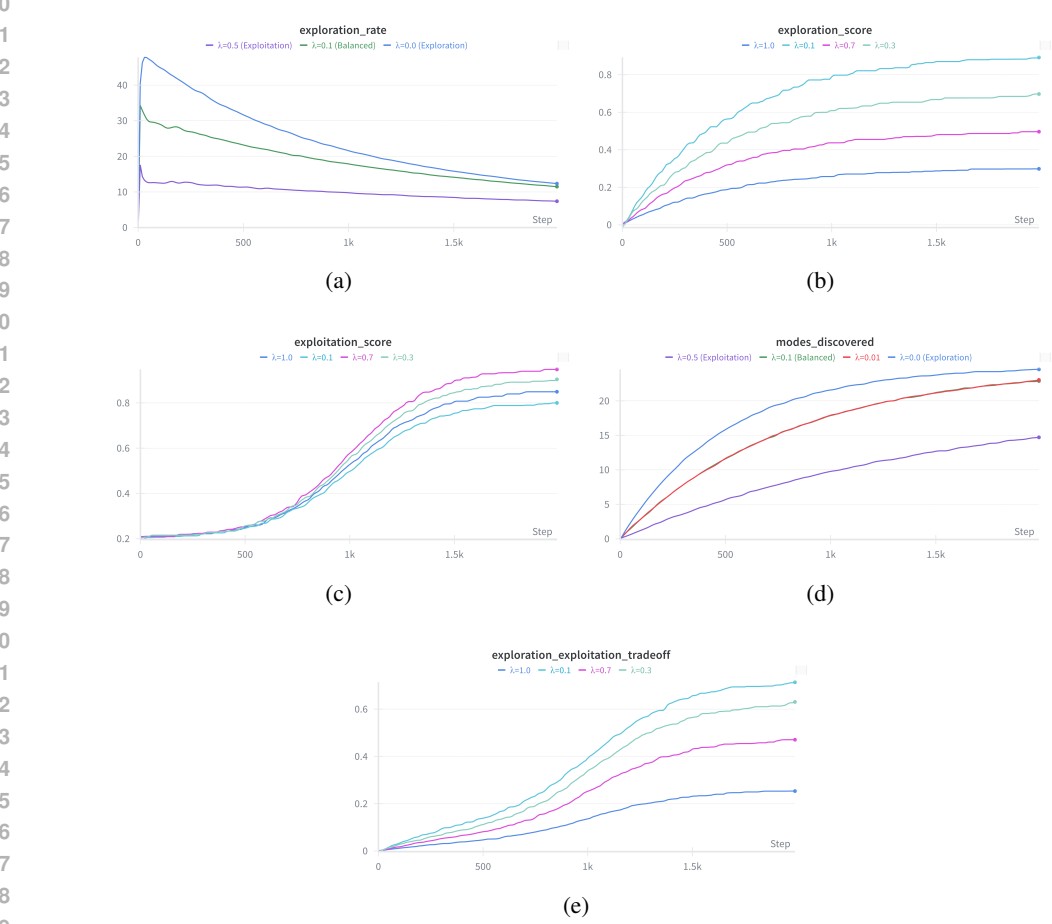

Figure 4: **(Hypergrid) Exploration-exploitation dynamics under teachability control.** (a) Exploration rate modulation through $\lambda$ parameter adjustment. (b) Exploration scores demonstrating DATE-GFN's superior discovery capabilities. (c) Exploitation efficiency showing effective utilization of discovered knowledge. (d) Mode discovery as a function of $\lambda$, revealing optimal parameter ranges. (e) Exploration-exploitation trade-off curves illustrating the principled control mechanism.

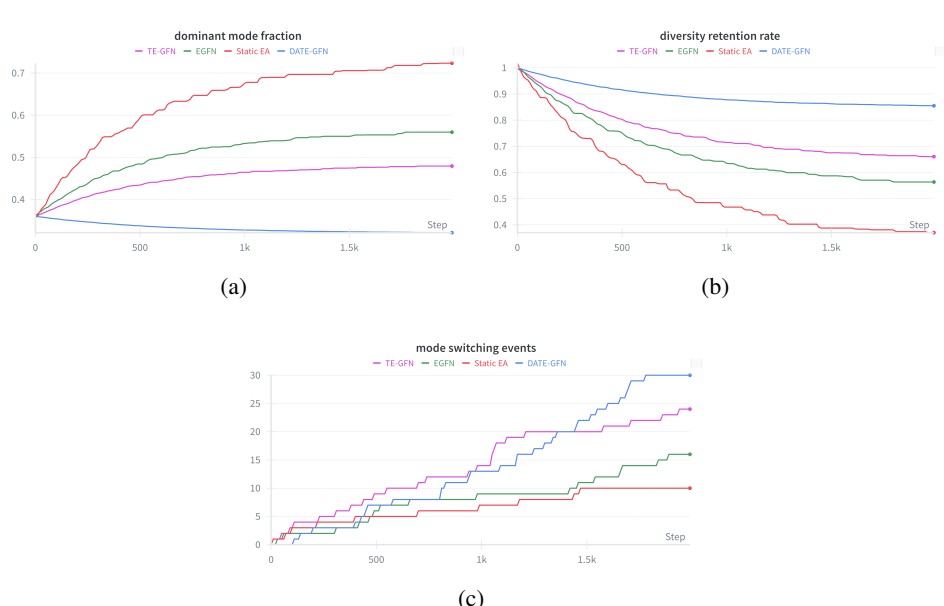

(a)

(b)

(c)

Figure 5: **(Hypergrid) Mode collapse mitigation through dynamic escape mechanisms.** (a) Dominant mode fraction over time showing DATE-GFN's resistance to premature convergence compared to baseline methods. (b) Diversity retention curves demonstrating 1.875× longer preservation of population heterogeneity through the theoretical escape condition.

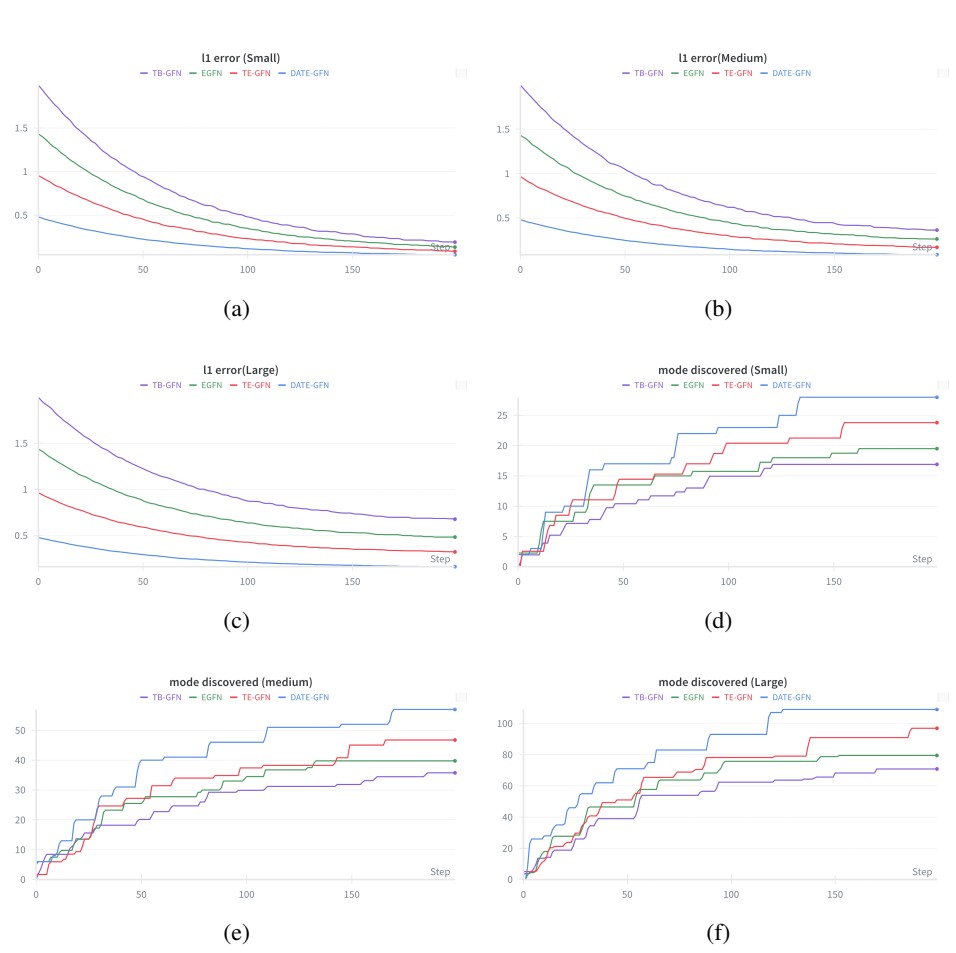

Figure 6: **(Hypergrid) Scalability analysis across problem complexities.** Top row: L1 error performance on (a) small, (b) medium, and (c) large problems showing consistent 76-79% improvements. Middle row: Mode discovery on (d) small, (e) medium, and (f) large problems demonstrating 54-65% advantages that increase with complexity.

## 11.5 SOLUBLE EPOXY HYDROLASE (sEH) BINDER GENERATION.

To demonstrate the scalability and practical utility of DATE-GFN, we adopt the challenging sEH binder generation task, a benchmark for *de novo* drug design. The objective is to generate novel, high-affinity molecules for the sEH protein, a significant therapeutic target. Following established protocols (Bengio et al. (2021a), Pan et al. (2023a)), molecules are constructed sequentially as graph structures from a vocabulary of 72 chemical blocks, using a junction tree modeling approach. This sequential process, with trajectories of up to 8 blocks, in a vast state space ($|\mathcal{X}| \approx 10^{16}$), combined with a sparse terminal reward signal—a proxy for binding energy—creates a severe long-horizon credit assignment problem. Furthermore, the task explicitly requires diverse solutions, with a 'mode' defined as a molecule with a reward $> 7.5$ and Tanimoto similarity $< 0.7$ to other modes. The state-of-the-art architecture for this task is a sophisticated Message Passing Neural Network (MPNN). This high-capacity model makes the *realization gap* a critical concern. Therefore, we instantiate our framework by using MPNN-based models for both the student GFlowNet and the population of critic value functions. This allows the evolutionary search for teachable, high-value critics to operate directly in the relevant function space of graph-based chemical intuition. This setup provides an ideal and rigorous testbed, as it simultaneously evaluates DATE-GFN's ability to solve the credit assignment problem, manage the realization gap in complex models, and enhance the core GFlowNet objective of diverse mode discovery.

Table 5: Performance scaling on the sEH binder task. Each row represents a method's performance at a specific maximum molecule size. The results clearly show DATE-GFN's superior performance, lower variance, and better mode discovery, with its advantages widening as problem complexity increases.

| Method | Max Atoms | Avg. Top-K Reward ↑ | Reward Std. Dev. ↓ | Modes Discovered (#) ↑ |
|---|---|---|---|---|
| GFN (TB) | 10 | $0.73 \pm 0.04$ | 0.08 | $125 \pm 15$ |
| EGFN | 10 | $0.78 \pm 0.03$ | 0.06 | $180 \pm 20$ |
| **DATE-GFN** | 10 | $\mathbf{0.85 \pm 0.02}$ | **0.03** | $\mathbf{270 \pm 25}$ |
| GFN (TB) | 15 | $0.68 \pm 0.05$ | 0.11 | $95 \pm 18$ |
| EGFN | 15 | $0.75 \pm 0.04$ | 0.09 | $155 \pm 22$ |
| **DATE-GFN** | 15 | $\mathbf{0.83 \pm 0.02}$ | **0.04** | $\mathbf{255 \pm 28}$ |
| GFN (TB) | 25 | $0.55 \pm 0.08$ | 0.19 | $35 \pm 15$ |
| EGFN | 25 | $0.66 \pm 0.06$ | 0.14 | $90 \pm 28$ |
| **DATE-GFN** | 25 | $\mathbf{0.78 \pm 0.03}$ | **0.06** | $\mathbf{210 \pm 32}$ |

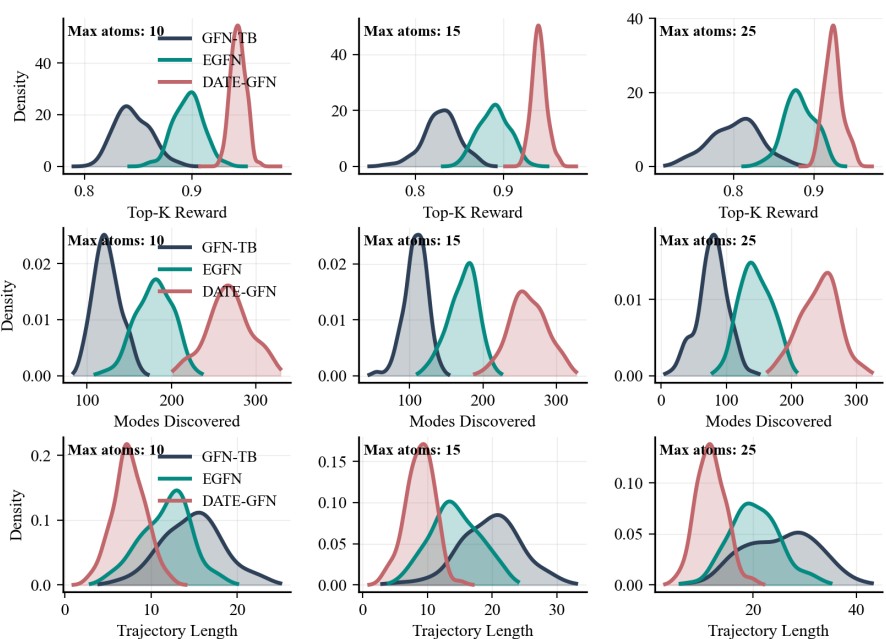

Figure 7: Comprehensive method comparison via kernel density estimation across molecular complexity levels. **Top row**: Top-K reward distributions showing DATE-GFN's superior performance with minimal variance across all complexity levels (10, 15, 25 max atoms). **Middle row**: Mode discovery distributions demonstrating DATE-GFN's ability to identify 2-3× more diverse molecular structures than baseline methods. **Bottom row**: Trajectory length distributions revealing DATE-GFN's computational efficiency, requiring 2-3× fewer generation steps while maintaining superior performance. DATE-GFN consistently dominates all three metrics with narrow, optimally-positioned distributions, validating its triple advantage of performance, diversity, and efficiency in the sEH binder optimization task.

