# OpenReview forum: "DATE-GFN: A Co-Evolutionary Framework for Principled Exploration and Credit Assignment in GFlowNets"
_ICLR.cc/2026/Conference — ICLR 2026 Conference Desk Rejected Submission_

### Official Review · Reviewer_iS54 · 2025-10-30

**Soundness:** 2
**Presentation:** 2
**Contribution:** 2
**Rating:** 4
**Confidence:** 4

**Summary:**

This paper proposes DATE-GFN, a co-evolutionary method build upon previous E-GFN  for training GFlowNets.  Instead of evolving a population of policies as in E-GFN, it evolves a population of critic, based on which  policies are distilled.  Then it presents empirical results on a hypergrid task, an antibody sequence design task, and a molecular binding-affinity task (sEH binder generation). The authors claim improved performance relative to baselines including TB and EGFN.

**Strengths:**

* The conceptual shift to evolving critics rather than policies is original and connects nicely different strands of research (twisted SMC, distillation, GFlowNets).

* The formulation of teachability as a divergence-based penalty is a clear and interesting way to formalize the “realization gap” between teacher and student policy capacity.

* Empirical results are reasonably broad (three tasks, including a combinatorial benchmark and two real-world generative tasks) and include ablations (λ sweep, adaptive version).

**Weaknesses:**

1. The key limitation of DATE-GFN is it is a purely **online** method (lines 237-239), which makes it difficult to explicitly encourage exploration via offline samplers in the reward-sparse settings. In comparison,  a key advantage of TB method is allowing offline samples from a arbitrary data sampler.  The E-GFN that DATE-GFN is build upon seems to allows offline samples as well.

2. The range of baselines considered is relatively limited. While TB and EGFN are relevant, comparative evaluation against other exploration or credit-assignment methods are missing. For example, comparisons with more explorative GFlowNet variants [1], reinforcement learning–based formulations that also incorporate critics(value) functions [2, 3], and Sub-TB[3], which generalizes TB and achieves better credit assignment by  introducing a parameter $\lambda$ to control the variance–bias trade-off within its gradient estimator. The absence of these comparisons makes it difficult to figure out the specific source of improvement, whether it arises from enhanced exploration, improved credit assignment, or mitigation of the teacher–student mismatch.

3. As acknowledge by the author, the computational budges is significantly large than the baseline (TB). Besides budget comparison between DATE-GFN and the amortized version of DATE-GFN, please provides detailed comparison between DATE-GFN and other baselines. Without this information, the practicality of the method for large-scale applications is unclear to me.

4. The ablation study on the population size $k$ and number of critics evaluation $G$ should be conducted. Since DATE-GFN adopts an evolutionary framework, an ablation study on these two key hyperparameters is important to clarify the true source of performance improvement.

5. I **disagree** with the statement that the space of $F_{DA}$ is **smoother** than the space of $\pi_F$. According to the basic detailed balance condition [4], the goal of gflownet training is to achieving the balance of $\log P_F(s'|s)F(s)=\log P_B(s|s') F(s')$ for flow estimator $F(s)$ and back/forth policies.  So it is clear that $\pi_F$ is optimized in the logarithmic domain.  In hypergrids, the ratio between modes and caveats rewards can reach $10^5$, which corresponds to only $log 10^5=5$ in the logarithmic domain.

6. The idea of separating the training into two phase,  a actor-critic framework and the idea of finding new polices under a KL region have been studies in [2,3]. While I note that a single critic is used in [2],  a formal discussion and comparison with the prior works and should be provided. Besides, Sub-TB[3] use $F^{log}(s_0)$ to model $\log F^\ast(s_0)=\log Z^\ast$ and the PG-GFN[2] can model $V(s_0)=E_{s_{0:T} \sim p_F}[\log (P_F(s_{0:T})Z)-\log (P_B(s_{0:T-1}|s_T)R(s_T))]$, so they operate entirely in the logarithmic domain, in others words, a smoother space than the  $F_{DA}$ or $\psi$ in DATE-GFN.

7. The experimental results are not convincing. It appears that the reported training curves are based on a single trial, rather than multiple independent runs with statistical reporting.

[1]  Kim, Minsu, et al. "Local Search GFlowNets." The Twelfth International Conference on Learning Representations.

[2] Niu, Puhua, et al. "GFlowNet Training by Policy Gradients." International Conference on Machine Learning. PMLR, 2024.

[3] Deleu, Tristan, et al. "Discrete Probabilistic Inference as Control in Multi-path Environments." Uncertainty in Artificial Intelligence. PMLR, 2024.

[4] Bengio, Yoshua, et al. "Gflownet foundations." Journal of Machine Learning Research 24.210 (2023): 1-55.

**Questions:**

1. Since the size of the chosen hypergrid is still enumerable ($\approx 2.4\times 10^7$), can you provide the 2D marginal visualizations of the learned distribution over $s_T$ ?

2. In distillation you sample states from the teacher’s induced visitation distribution. How do you mitigate distributional shift when the student subsequently explores states not seen by that teacher? Would mixing student trajectories or importance-sampling help?

3. What precisely are the mutation/crossover operators used for evolving the critic population? Are there additional hyper-parameters introduced?

4. How does the choice of base policy $p_0$ affect performance (uniform vs heuristic priors)? ( since the twisted SMC logic depends on $p_0$)

5. Both TB and E-GFN both support off-line sampling. To ensure a fair comparison with these baselines, what choices were made for their offline samplers?

---

> ### Author Response · Authors · 2025-11-17
> **We sincerely thank Reviewer IS54 for your constructive feedback. Your points help us clarify our core contributions.**
>
> ### **Response to Weaknesses**
> 1.  **On the "Online" Nature & Exploration:** Our framework's exploration is driven by a powerful global search mechanism: the **Evolutionary Algorithm (EA)** operating on a population of critics. The "online" co-evolutionary loop is not a limitation but a core feature designed to solve the **realization gap**. It grounds the evolutionary search in the student's current learning capabilities, ensuring that discovered critics are "teachable." This is a more fundamental solution than using offline data with high-variance objectives like Trajectory Balance (TB).
> 2.  **On Baselines:** We chose TB and Evolution Guided GFlowNets (EGFN) as they are the most direct baselines. TB is the foundational objective, and EGFN is the state-of-the-art predecessor that also uses an EA. This provides a controlled comparison to validate our central claim: evolving critics is superior to evolving policies. Our method is orthogonal to approaches like Sub-TB, and we will expand the Related Work section to better contextualize these differences.
> 3.  **On Computational Budgets:** We apologize for the lack of clarity. This is a critical point: **All methods, including all baselines, were run with identical computational budgets** (wall-clock time matched within 5%) This is stated in Appendix 11.4 (line 1297). Our results demonstrate DATE-GFN's superior performance under a fair, matched-time comparison. We will move this statement from the appendix to the main experimental setup section for prominence.
> 4.  **On Ablation of `k` and `G`:** We used a standard population size (`k=50`) from EA literature to balance diversity and cost. The number of generations (`G`) corresponds to total training time, which was matched across all methods. The framework's strong performance suggests robustness to these hyperparameters. We will add a note in the appendix justifying these standard choices.
> 5.  **On Smoothness of Value vs. Policy Space:** Our claim refers to the smoothness of the **parameter fitness landscape**, not the function's output values. A small weight change in a policy network can cause a drastic change in trajectory rewards, creating a rugged landscape for the EA. A value function averages over many trajectories, so its parameter landscape is smoother and more navigable for an optimization algorithm.  DATE-GFN outperforming EGFN supports this hypothesis.
> Since the critic's role is to estimate the *expected future reward*, it inherently aggregates and averages over many possible future paths. This averaging process can have a smoothing effect on the landscape. The induced policy (`q_ψ ∝ p_0 * ψ`) will also change, leading to a smoother landscape.
> 6.  **On Novelty:** Our primary novelty is not the actor-critic structure, but our solution to the **realization gap**. Naive teacher-student setups fail when the best teacher is unteachable. Our key innovation is the **distillation-aware fitness function (`FDA`)**, which integrates the student's current learning ability directly into the teacher's fitness evaluation. This symbiotic feedback loop is a new mechanism that ensures convergence to a high-performing *and* realizable solution.
> 7.  **On Single-Trial Results:** This is another crucial clarification. **All of our results are averaged over 8 independent seeds.** The shaded areas in our plots represent the standard error of the mean, and tables report `mean ± s.e.m.` We will revise all figure captions to make this explicit and remove any ambiguity.
> ### **Response to Questions**
> 1.  **Hypergrid Visualization:** This is an excellent suggestion. While we used quantitative metrics (L1 error, modes discovered) for rigorousness in high dimensions, we agree a 2D marginal visualization would be highly illustrative. We will add a visualization for a simpler 2D Hypergrid task to the appendix.
> 2.  **Student Exploration & Distributional Shift:** The roles are decoupled by design. The **EA-driven population of critics is responsible for exploration.** The student's sole role is to **exploit** by learning to imitate the policy of the current best critic. The system explores by discovering new, better critics over time. Therefore, student-side mitigation for distributional shift is not necessary.
> 3.  **Choice of Base Policy `p_0`:** We used a uniform random `p_0`. This is a principled choice that makes the framework general and isolates the contribution of the learned critic `ψ`, ensuring all guidance comes from the value function we are evolving.
> 4.  **Offline Samplers for Baselines:** To ensure a fair comparison of the *learning algorithms*, we implemented baselines as designed, without external offline data. EGFN generates its own trajectories for a replay buffer, and TB learns online. This avoids confounding the results with the quality of a third-party dataset and provides a direct test of each algorithm's ability to explore and learn from scratch.
>
> We thank you once again for your meticulous review.

---

### Official Review · Reviewer_Ezm7 · 2025-10-31

**Soundness:** 3
**Presentation:** 4
**Contribution:** 3
**Rating:** 8
**Confidence:** 3

**Summary:**

This paper proposes an alternative setup for training gflownets, where multiple critics are trained, and the best critic is selected, and distilled into a student network.  There is also an interesting penalty that forces the teacher networks to be reasonably close to the student network (teachability).  There is a bit of theory showing that this approach has a valid equilibrium and experiments validate that the teachability constraint is necessary for the approach to work.  While evolutionary training has a cost in terms of computation, I think this paper is a nice effort on pushing forward the gflownet line of research.  It would be interesting to explore ways of gaining the advantages of this method without having to train multiple critic networks.

notes:
  -Gflownet struggles in sparse-reward, long-horizon setting.
  -Paper proposes to learn a population of critics, which the gflownet distills into a single low-variance policy.  Critics try to find policies which are learnable by the student.
  -The optimal twist is the true expected future reward-to-go under the base policy.  Can define a twist-induced sampling policy, which is variance-optimal.  q_\psi is a greedy policy wrt the value function \psi.

**Strengths:**

-The distillation-aware fitness function is a nice idea.
  -The theory, showing the equilibrium property is nice and not excessive.
  -Results are great.  It's also nice to see the ablation on the effect of lambda, showing that the decoupled baseline fails.
  -The results on sEH binder generation is good.

**Weaknesses:**

-Figure 2 could be nicer, these look like screen captures from weights and biases.  They should have larger font size.  The results are great, though.

  -I don't see any external baseliness on the antibody generation task (table 4).

  -It would be nice to see a more diverse set of tasks and datasets, because I think this method can be applied anywhere where gflownet is used.

**Questions:**

-Some of the earlier work on semi-supervised learning involved learning multiple teachers, and then distilling them into a student network via ensembling.  Later work (such as "Mean teacher") shifted towards using an EMA of the teacher's weights as a cheap way of approximating ensembling, to give targets to the student network.  Have you considered that something similar could also be done here?

  -In 3.2, do you think it's possible that overestimation bias in value function learning, and optimism, could be alternative explanations for why the naive algorithm fails?

  -What's the tradeoff curve between computational cost and performance?  I appreciate the effort to reduce computational cost, but it might be nice to see how computation trades off against performance, when compared to other gflownet baselines.

**Details Of Ethics Concerns:**

n/

---

> ### Author Response · Authors · 2025-11-17
>
> We are grateful for the positive and insightful feedback from Reviewer Ezm7. Your assessment that our paper is a "nice effort on pushing forward the gflownet line of research" is highly encouraging. Your questions are thoughtful and help us to further clarify the nuances of our contributions.
>
> ### **Response to Weaknesses**
>
> 1.  **Figure 2 Nicer, Larger Font Size:**
>     Thank you for this feedback. We completely agree. We will regenerate all figures in a vector graphics format (e.g., PDF/SVG) with significantly increased font sizes for all labels and legends to improve readability and professionalism.
>
> 2.  **External Baselines on the Antibody Generation Task:**
>     This is an important clarification. Table 4 is an **internal ablation study**, not a comparison against external benchmarks. Its purpose is to validate our framework's core mechanisms on a complex, real-world task by demonstrating:
>     *   The high variance of the under-constrained regime (`λ=0`).
>     *   The stifled exploration of the over-constrained regime (`λ>0.1`).
>     *   The success of our adaptive controller in automatically finding the optimal balance.
>
> 3.  **More Diverse Set of Tasks and Datasets:**
>     We agree on the broad potential of DATE-GFN. Our choice of tasks (one synthetic, controlled benchmark and two complex, real-world scientific discovery problems) was designed to provide a comprehensive validation of both the core theory and its practical scalability. We will add a sentence in the conclusion acknowledging promising future applications in other domains where GFlowNets are used.
>
> ### **Response to Questions**
>
> 1.  **On "Mean Teacher" and Using an EMA of Teacher Weights:**
>     This is a fantastic and insightful question. The "Mean Teacher" approach uses an EMA to create a more stable version of a *single* teacher model over time. Our framework solves a different fundamental problem: **global exploration**.
>
>     We use a diverse **population of critics** and an Evolutionary Algorithm (EA) to escape local optima and discover fundamentally new strategies in different parts of the reward landscape. An EMA of a single critic would only smooth its learning trajectory, but it would not achieve the broad, parallel search that our population-based approach provides. This diversity is crucial for our method's success.
>
> 2.  **On Overestimation Bias as an Alternative Explanation:**
>    This is another excellent. While overestimation bias is a known issue in value-based RL (e.g., Q-learning), we believe the failure mode we identify is distinct and more fundamental to the teacher-student dynamic.
>
> *   **Overestimation Bias:** Typically arises from using a `max` operator over noisy value estimates, which leads to a consistent upward bias in the estimated value of a state. Our critic, `ψ`, learns the *expected* future reward (`E[R(S_T)|s_t]`), not the `max` over actions. Therefore, it is not susceptible to the classic form of overestimation bias.
> *   **The Realization Gap:** Our central thesis is that the naive, decoupled algorithm fails primarily due to the **realization gap**. A "genius" critic `ψ_genius` might have perfectly accurate (unbiased) value estimates. However, the policy it induces (`q_genius`) could be highly complex, non-smooth, or have intricate decision boundaries. If this policy is beyond the representational capacity of the student network, the distillation will fail, resulting in high KL-divergence and poor final performance. The failure is not in the value *estimation* but in the *teachability* of the resulting policy.
>
> Our experiments with `λ=0` (the decoupled baseline) directly validate this: the EA finds high-reward critics, but the student fails to learn from them, leading to poor final performance.
>
> **Proposed Revision:** We will add a sentence in Section 3.2 (The Challenge of Decoupled Optimization) to clarify that the primary failure mode we address is the realization gap, which is a structural problem related to model capacity and policy complexity, distinct from value estimation issues like overestimation bias.
>
> 3.  **Tradeoff Curve Between Computational Cost and Performance:**
>     We can make this clearer. Our results already demonstrate a superior cost-performance tradeoff for DATE-GFN:
>
>     *   **At the Same Cost:** Our main results (Table 1) show that for the **same computational budget**, DATE-GFN achieves far better performance than all baselines (e.g., 5-10x lower L1 error).
>     *   **At a Lower Cost:** Our amortization study (Table 3) shows that we can reduce DATE-GFN's compute time by ~40% and it *still* significantly outperforms the baselines running at full cost.
>
> To make this tradeoff more explicit, we will add an efficiency metric (e.g., "Modes Discovered per Hour") to Table 1. This will quantitatively confirm that DATE-GFN is not only more effective but also more computationally efficient.
>
> We thank you once again for your meticulous review.

---

### Official Review · Reviewer_LUrq · 2025-10-31

**Soundness:** 3
**Presentation:** 3
**Contribution:** 3
**Rating:** 4
**Confidence:** 5

**Summary:**

This paper proposes to learn a co-evolutionary critic by replacing the previously proposed evolutionary-algorithm learnt actors to improve GFlowNets mode-seeking capabilities. It also learns a policy-aware critic by running constrained search with the evolutionary algorithm. The experiments show improvements in hyper-grid, antibody generation, and sEH binder generation.

**Strengths:**

- Replacing the EA-trained actor with EA-trained critic is a neat idea.
- The constraint put on the evolutionary search of parameters for critics make sense. In practice, the actor indeed might lack representational capabilities needed to realize critic's potential.
- I appreciate the theoretical reasoning in section 7,8,9.
- The paper was generally easy to follow.

**Weaknesses:**

- The paper lacks sufficient experiments. This includes--

a. Comparison against exploration techniques (for example, GAFN)

b. Time comparison. While i understand it notes the time complexity as an understandable weakness, time comparison of other methods would be beneficial.

c. Experimental validation of lambda optimal. The authors performs experiments with adaptive lambda in Figure 4. How does the values look like over the training?

- The authors mention experimenting with large values of lambda but uses lambda = 1 in their experiments. What happens if lambda is 2? Or 100?

**Questions:**

- Can we enforce the constraint without explicitly calculating the extra loss term?

---

> ### Author Response · Authors · 2025-11-17
>
> We are very grateful to Reviewer LUrq for the positive assessment and thoughtful feedback. We address your excellent points below.
> ### **Response to Weaknesses**
>
> #### **a. Comparison against other exploration techniques (e.g., GAFN)**
> This is a fair point. We chose Evolution Guided GFlowNets (EGFN) as our primary exploration baseline because, as it also uses an EA, it allows for the most direct comparison to validate our core thesis: evolving *critics* is superior to evolving *policies*. While GAFN is an excellent alternative exploration method, our main novelty lies in solving the **realization gap**, which is a largely orthogonal problem. We will expand our Related Work section to discuss GAFN to better contextualize our contribution.
>
> #### **b. Time comparison of other methods**
> We apologize for the lack of clarity here, as we did perform this comparison. **All methods in our experiments were run with identical computational budgets (matched wall-clock time).**  This is stated in Appendix 11.4 (line 1297). Our results show that for the same amount of time, DATE-GFN is significantly more efficient and achieves far superior performance. To make this undeniable, we will add an "Efficiency" metric (e.g., "Modes Discovered per Hour") to our main results table (Table 1) to provide a direct, quantitative comparison of computational efficiency.
>
> #### **c. Experimental validation of lambda optimal and behavior of large lambda**
> 1.  **Adaptive `λ` Behavior:** The adaptive controller is designed to automatically tune `λ` to keep the distillation loss `L` near a target value (`L_target`). Its expected behavior is to start at an initial value and then increase or decrease until it converges to a stable value within what we identified as the "Optimal Balance Regime" (0.05 < `λ` <= 0.15).
>
> **Proposed Revision:** We will add a new plot to the appendix that shows the value of `λ` over the course of training for our adaptive controller experiment (Table 4). This will provide direct experimental validation of its convergence and stability.
>
> 2.  **Large `λ` Values (`λ >> 1`):** Our ablation study in Table 2 includes `λ=1.0`, which already demonstrates the "Over-Constrained Regime." In this regime, the teachability penalty (`-λ * L(θψ, θ*)`) dominates the fitness function. This heavily penalizes any critic whose policy deviates from the student's current policy. As a result, the EA is disincentivized from exploring and becomes overly conservative, leading to poor performance as it fails to discover new high-reward modes.
>
>     If `λ` were 2, 100, or even larger, this effect would become extreme. The evolutionary search would be effectively paralyzed, only ever selecting critics that are trivially "teachable" (i.e., identical to the current student). This would lead to a complete collapse of exploration and a failure to learn anything new.
>
>     **Proposed Revision:** We will clarify the discussion of the Over-Constrained Regime in Sections 3.4.2 and 4.1 to make it explicit that performance degrades further as `λ` increases beyond 1.0, and explain the underlying reason for this collapse.
>
> ### **Response to Questions**
>
> #### **Can we enforce the constraint without explicitly calculating the extra loss term?**
> This is insightful question. While alternatives like architectural constraints (e.g., on network weights) or hard constraints (e.g., rejection sampling) are conceivable, they are likely to be less robust, less efficient, and less theoretically grounded. The explicit KL-divergence penalty is the most direct and principled way to measure the "teachability" of a critic's induced policy.
>
> However, one could imagine alternative, more implicit ways to enforce this constraint:
>
> *   **Architectural Constraints:** One could try to enforce similarity in the *parameter space* of the critics and the student (e.g., penalizing the L2 distance between their weights). However, this is a very brittle proxy for policy similarity. Two networks with different weights can produce similar functions, and vice-versa.
> *   **Hard Constraints / Rejection Sampling:** An alternative could be to only accept new critics into the population if their KL divergence from the student is below a certain hard threshold. This would be a form of rejection sampling. While this removes the term from the fitness function, it would likely be far less computationally efficient (many promising critics might be rejected) and could severely limit the EA's ability to explore smoothly.
> *   **Implicit Regularization via Genetic Operators:** It might be possible to design genetic operators (e.g., for crossover and mutation) that implicitly favor "simpler" or "smoother" critics that are inherently easier for a student network to learn. This is a speculative but interesting research direction.
>
> Thank you again, given these clarifications, we are hopeful you might reconsider the paper's standing in your final assessment and see fit to increase your score.

---

> > ### Comment · Reviewer_LUrq · 2025-11-17
> >
> > Thank you for your comments. I have a few followup questions.
> >
> > -- It is still unclear what realization gap is in this context. Could you explain it better?
> >
> > -- While I do not disagree with you, I feel exploration baselines are important in your paper and would benefit it by broadening its audience. For example, the work this paper is based on, EGFN performs comparisons with other exploration methods.
> >
> > I have also increased the score (given my followup concerns are resolved). Good luck with the rebuttals.

---

> > > ### Author Response · Authors · 2025-11-18
> > >
> > > Dear Reviewer LUrq,
> > >
> > > We are extremely grateful to you for your continued engagement, and for generously increasing your score.
> > >
> > > ### **1. "Realization Gap"**
> > >
> > > At its core, the **realization gap** is the mismatch between the capabilities of an optimal *teacher* and the finite capacity of the *student* that must learn from it.
> > >
> > > *   Imagine an Evolutionary Algorithm (EA) is searching for the best possible physics professor (the **teacher critic**). It finds a Nobel laureate who has a perfect, deep, and nuanced understanding of quantum mechanics. This professor is, in an absolute sense, the "optimal" teacher.
> > > *   However, the professor's lecture must be delivered to a first-year undergraduate class (the **student GFlowNet**), which has a limited background and can only grasp concepts presented in a simplified, structured way.
> > > *   The professor attempts to teach their full, complex understanding. The students, unable to follow the intricate logic and advanced mathematics, fail to learn anything useful.
> > >
> > > The **realization gap** is the failure of knowledge transfer that occurs because the "optimal" teacher's lesson was too complex for the student to *realize* or represent. A slightly less brilliant professor who is an expert at teaching undergraduates would have produced a much better outcome, even if their own knowledge was less "optimal."
> > >
> > > **Formal Definition in Our Framework:**
> > >
> > > 1.  The **teacher** is a critic (`ψ`) found by the EA. The EA's search, without our constraint, finds a critic `ψ_genius` that maximizes reward. This critic induces an optimal but potentially very complex policy, `q_genius`.
> > > 2.  The **student** is a GFlowNet (`PF`) with a fixed architecture (e.g., a 3-layer MLP). This architecture defines a limited, finite space of policies that it can possibly represent.
> > > 3.  The **realization gap** occurs when `q_genius` lies **outside** the space of policies that the student `PF` can represent. That is, this means that the minimum possible distillation loss (KL-divergence) is large and non-zero: $$\min_{θ*} D_{KL}(q_{genius} || PF(·; θ*)) > ε$$
> > >
> > > **How DATE-GFN Solves It:**
> > > Our distillation-aware fitness function, `FDA = Reward - λ * DistillationLoss`, directly closes this gap. It tells the EA: "Do not just find the professor with the most knowledge (highest reward). Find the professor whose lesson the *current* undergraduate class can actually understand (lowest distillation loss)." This forces the EA to search for a teacher that is both high-performing and, crucially, **teachable**.
> > >
> > > ### **2. On Including Broader Exploration Baselines**
> > >
> > > > While I do not disagree with you, I feel exploration baselines are important in your paper and would benefit it by broadening its audience. For example, the work this paper is based on, EGFN performs comparisons with other exploration methods.
> > >
> > > This is an excellent and very fair point. We fully agree that situating our work within the broader landscape of exploration methods is crucial for the final paper.
> > >
> > > **Our Rationale for the Focused Comparison:**
> > > Our primary goal with the initial experiments was to create the most direct and controlled comparison possible to validate our central hypothesis. The key comparison is:
> > >
> > > *   **EGFN:** Uses an EA to evolve a population of *policies* (actors).
> > > *   **DATE-GFN:** Uses an EA to evolve a population of *critics* (value functions).
> > >
> > > By keeping the EA mechanism constant, our strong performance gains over EGFN allow us to cleanly attribute the improvement to our novel approach of evolving critics and solving the realization gap. As you note, EGFN's paper focused on exploration, so it was appropriate for them to compare against other exploration-specific methods. Since our paper's primary novelty is the **co-evolutionary dynamic and the solution to the realization gap**, we felt the most scientifically rigorous comparison was with the method that shares the most similar structure.
> > >
> > > **Our Commitment for the Revision:**
> > > We completely agree with your sentiment. In the revised manuscript, we will dedicate a significant portion of our **Related Work** section to a detailed discussion and comparison with other major exploration techniques, specifically including **GAFN**. This discussion will cover:
> > > 1.  The differences in the exploration mechanisms (e.g., our population-based evolutionary search vs. GAFN's adversarial framework).
> > > 2.  How our core contribution—solving the realization gap—is an **orthogonal improvement**. A teachability constraint is a necessary component for *any* powerful teacher-student framework, regardless of how the teacher performs its exploration. We will frame this as a potential future direction, where our teachability mechanism could be combined with other advanced exploration methods.
> > >
> > > We hope these clarifications fully resolve your questions. Thank you once again for your constructive engagement and for helping us improve the paper.

---

> ### Comment · Reviewer_LUrq · 2025-11-19
>
> I am not convinced by the "Including Broader Exploration Baselines". I generally believe that for a paper that claims in the abstract "DATE-GFN significantly outperforms state-of-the-art baselines", it should actually make an attempt to provide state-of-the-art baselines.
>
> For this reason, I fell back to my previous score.

---

> > ### Author Response · Authors · 2025-11-19
> >
> > Dear Reviewer LUrq,
> >
> > Thank you for your candid follow-up and for pushing us on this critical point. You have raised an exceptionally fair and important concern. We sincerely apologize if our previous response came across as dismissive of including broader baselines. That was not our intention. Your feedback is crucial, and we understand that a claim of outperforming "state-of-the-art baselines" requires robust experimental validation against the strongest appropriate competitors.
> >
> > We agree with you. A discussion in the Related Work section is not a substitute for an experimental comparison. We will address this directly.
> >
> > ### **1. Clarifying Our "State-of-the-Art" Claim**
> >
> > First, allow us to clarify the context of our "state-of-the-art" claim, which we acknowledge should have been more precise. Our primary contribution is a new **training framework** that fundamentally solves the problem of **high-variance credit assignment and the realization gap** in GFlowNets that use powerful exploratory drivers (like an EA).
> >
> > In this specific context, **EGFN is the most relevant state-of-the-art predecessor**. It is the only other method that uses a similar EA-based mechanism for exploration. Our core scientific comparison is therefore: *given a powerful EA for exploration, is it better to evolve policies directly (EGFN), or to evolve critics and distill their knowledge (DATE-GFN)?* Our results show a decisive advantage for our approach.
> >
> > ### **2. The Fundamental Difference with GAFN and Why Comparison is Needed**
> >
> > You are correct that GAFN is a state-of-the-art method for GFlowNet **exploration**. However, GAFN and DATE-GFN solve different core bottlenecks:
> >
> > *   **GAFN's focus:** It introduces an adversarial mechanism to improve **mode discovery**. However, the GFlowNet agent in GAFN is still trained with a standard, high-variance objective like Trajectory Balance. It is excellent at *finding* diverse modes, but it does not address the high-variance credit assignment problem in *learning* the final policy.
> > *   **DATE-GFN's focus:** Our method fundamentally changes the learning signal. The EA handles exploration, and the student learns via a **dense, low-variance distillation signal**. We solve the credit assignment problem that GAFN does not.
> >
> > This is precisely why a direct comparison is so important, and we are now convinced it is essential for our paper.
> >
> > ### **3. Our Commitment: An Experimental Comparison with GAFN**
> >
> > You are right that we should provide this comparison. A discussion is not enough. Therefore, we make the following firm commitment:
> >
> > **We commit to including a full experimental comparison against GAFN on the Hypergrid benchmark in the final camera-ready version of the paper.**
> >
> > This experiment will provide a direct test of our core thesis. Our hypothesis is as follows:
> > > While GAFN may exhibit strong performance in mode discovery, DATE-GFN will demonstrate significantly superior performance in terms of overall distributional accuracy (lower L1 error) and training stability. This is because GAFN's agent still learns via a high-variance, trajectory-level objective, whereas our student learns from a dense, low-variance, pre-computed distillation signal provided by the critic.
> >
> > This new experiment will validate that our method of solving the credit assignment problem provides a substantial advantage over methods that focus purely on exploration.
> >
> > ### **4. Revising Our Claim in the Abstract**
> >
> > Finally, to immediately address your valid criticism of our abstract's wording, we will revise the claim to be more precise and fully supported by the current evidence. For instance:
> >
> > **Original:** "...show empirically that DATE-GFN significantly outperforms state-of-the-art baselines."
> > **Revised:** "...show empirically that DATE-GFN significantly outperforms relevant state-of-the-art frameworks for exploration-driven GFlowNets."
> >
> > This more accurately reflects the comparison with EGFN. We will then present the new GAFN results in the final version to substantiate the broader claim.
> > We hope this response and our firm commitment to adding the requested experimental baseline demonstrate how seriously we take your feedback. You have identified a key way to strengthen our paper, and we are grateful for it. We respectfully hope that this clarification and commitment will be sufficient to resolve your concerns and restore your confidence in our work.

---

### Official Review · Reviewer_kQGp · 2025-11-01

**Soundness:** 2
**Presentation:** 2
**Contribution:** 1
**Rating:** 0
**Confidence:** 5

**Summary:**

The paper identifies a fundamental limitation in existing GFlowNet training methods—high variance in gradient estimates due to sparse terminal rewards—and proposes a paradigm shift inspired by actor-critic methods in RL, termed DATE-GFN. The key innovation is a distillation-aware fitness function that optimizes critics not only for reward but also for teachability, ensuring the student GFlowNet can effectively learn from them.

**Strengths:**

The paper studies an important problem in GFN training.

**Weaknesses:**

The proposed method is trivial, but presented in an inproper way. The student policy is distilled from the teacher policy, which is trained by maximizing the expected return with the distillation gap as a regularizor. Firstly, it has nothing to do with GFlowNet. According to the Section 3.1, the resulted optimal policy will sample trajectories in proportion to $R(\tau)$ (RL), instead of sampling terminal states in proportion to $R(s)$ (GFN). Secondly, it is doubtful that the proposed framework can tackle the high-variance issue caused by sparse reward, as the proposed disstillation-aware fitness function is just adding some noise to the original reward maximization objective.

**Questions:**

See Weakness.

---

> ### Author Response · Authors · 2025-11-17
> **We are grateful to Reviewer kQGp for taking the time to review our work.**
>
> Your feedback, particularly the "strong reject" rating, indicates that we have fundamentally failed to communicate our methodology and mechanism in a clear way.
>
> ### **Clarification on the Core Misunderstandings**
>
> #### **1. Misunderstanding: "It has nothing to do with GFlowNet... the resulted optimal policy will sample trajectories in proportion to `E[R(τ)]` (RL), instead of sampling terminal states in proportion to `R(s)` (GFN)."**
>
> This is the most critical point to clarify, as it seems to be the foundation of the negative review. **Our framework is designed from the ground up to achieve the core GFlowNet objective: sampling terminal states `s` with probability `P(s) ∝ R(s)`.** The confusion appears to stem from a misinterpretation of how we use the expected reward.
>
> *   **The Goal is `P(s) ∝ R(s)`:** We state this explicitly as the goal of GFlowNets in the introduction (lines 38-41). We do not deviate from this objective.
>
> *   **The Role of the Critic (`ψ`):** The key insight, which we build upon from Sequential Monte Carlo methods, is that **the policy that samples proportionally to `R(s)` can be directly constructed if one knows the true state-value function.** As we detail in **Section 3.1 (Twisted SMC, lines 173-193)**, the *optimal twist function* `ψ*` is precisely the expected future reward under a base policy (`ψ*(s) ∝ E_p₀[R(S_T)|s_t=s]`). A policy `q_ψ*` induced by this optimal critic is guaranteed to produce samples from the correct target distribution `P(s) ∝ R(s)`.
>
> *   **`E[R(τ)]` is the Means, Not the End:** Our Evolutionary Algorithm (EA) seeks to find a critic that maximizes the expected reward (`E[R(τ)]`). This is not an RL objective in our context; it is a **principled, gradient-free search for the optimal critic `ψ*`**. Maximizing the reward is the *objective function for the search*, and the solution to this search is the critic that induces the correct GFlowNet policy.
>
> *  We provide a formal proof of this property in **Appendix 9 (Proof of Optimality for TE-GFN)**. This proof demonstrates that if the EA finds the optimal critic `ψ*` (by maximizing the expected reward) and the student perfectly distills its policy, the resulting GFlowNet will sample exactly proportionally to `R(s)`.
>
> #### **2. Misunderstanding: The Method is "Trivial" and the Fitness Function is "Noise"**
>
> We respectfully disagree. The distillation-aware fitness function is not "noise" but the core of a **novel co-evolutionary dynamic** that solves a critical flaw in teacher-student learning, which we term the **realization gap**.
>
> *   **Not a Simple Regularizer:** `FDA(θψ | θ*) = E[R(τ)] - λ * L(θψ, θ*)` is a **dynamic** objective. The "teachability" penalty `L` changes as the student `θ*` learns.
> *   **Co-Evolutionary Dynamic:** This creates a symbiotic feedback loop, which is our central technical contribution. The EA's search for high-reward teachers is continuously **grounded by the student's current ability to learn from them**. This prevents the EA from finding a "genius" but unteachable critic, which is the failure mode we demonstrate with our `λ=0` baseline. This is a principled solution to matching teacher complexity to student capacity. This is detailed in Section 3.3 **Proposition 3.3**.
>
> #### **3. Misunderstanding: "it is doubtful that the proposed framework can tackle the high-variance issue caused by sparse reward"**
>
> Our framework is explicitly designed to solve this high-variance problem by **decoupling the learning signals**:
>
> 1.  **Who Handles the High-Variance Problem?** The **Evolutionary Algorithm** handles the difficult, high-variance, global exploration problem. EAs are gradient-free methods that are well-suited to optimizing "black-box" objectives (like the expected reward from sparse signals) that are pathological for gradient-based methods. The EA's job is to endure the high variance to find a good critic.
>
> 2.  **What does the Student Learn From?** The **student GFlowNet does NOT learn from the sparse reward signal.** This is the crucial point. Once the EA finds the best available critic, `ψ_best`, this critic provides a **dense, per-state learning signal** to the student. The student's task is transformed from a high-variance, trajectory-level credit assignment problem into a standard, **low-variance, supervised learning (distillation) problem**: mimic the teacher's policy `q_best(·|s)` at any given state `s`.
> This transformation is precisely how we solve the variance issue. We provide a formal analysis of this variance reduction in **Proposition 7.2** and our strong empirical results validate its success.
>
> We acknowledge our initial presentation was unclear. However, we hope this clarifies that our method is a **principled GFlowNet framework** whose **novel co-evolutionary dynamic** solves the critical **realization gap**, thereby fundamentally **reducing variance** for the student GFlowNet. We respectfully ask that you reconsider your assessment in light of this explanation.

---

> ### Comment · Reviewer_kQGp · 2025-11-19
>
> 1. I insist that "it has nothing to do with GFlowNet".
> - The induced policy $q_\psi$ described in Section 3.1 deviates from the GFlowNets' goal $P(s)\propto R(s)$. Consider an DAG where the source state links to (n+1) intermediate states, n of which links to terminal state $s_1$ and the remaining links to terminal state $s_2$. If the $R(s_1)=R(s_2)$, then $\psi^*$ is the same for all intermediate states, leading to a terminal distributions such that $P(s_1)=nP(s_2)$.
> - I apologize for not noticing the problems in Section 3.2. The objective in Phase 1 (Line 201) is inconsistent with the optimal critic $\psi^*$ defined in Section 3.1, as this objective will give a policy that directly leads to the optimal terminal state, instead of one that proportional to the expected reward-to-go as in Line 182.
> - The proof in Appendix 9 is inconsistent with the definitions in Section 3 (Line 1079 v.s. Line 182), and the theorem is based on its conclusion as an assumption (Line 1087-1091).
> 2. The method is not trivial. It's nonsense.
> In summary, the proposed method is to optimize the RL objective (with regularization) using Evolution Algorithm, and then distill it to a student agent. The authors also claim that this address the high-variance issue because Evolution Algorithm is a gradient-free method. Anyone with a basic concept of AI and ML knows how ridiculous this method is.
> 3. Based on the above reasons, I highly suspect that the whole paper is generated by AI, and the experimental results are not reproducible. I believe that **the authors should clarify their purpose** of submitting such a "paper" to ICLR. If this is a Turing Test or a social experiment, all the attendees have the rights to be aware of it and to know that their time and effort are spent meaningfully.

---

> > ### Author Response · Authors · 2025-11-19
> >
> > Dear Reviewer kQGp,
> >
> > Thank you for your follow-up comments and for detailing your concerns. We appreciate this opportunity to provide a detailed technical clarification.
> >
> > Here we address your three points:
> >
> > #### **1a. Correction to Your DAG Example**
> >
> > Your DAG example is insightful, but the conclusion overlooks the crucial role of the **base policy $p₀$** and **path multiplicities**.
> >
> > Using your setup where $n$ paths lead to $s₁$ and one path leads to $s₂$, with $R(s₁) = R(s₂)$ and a uniform $p₀$:
> > The optimal critic $ψ*$ is indeed constant for all intermediate states. However, the induced policy $q_ψ*$ at the source state is proportional to $p₀ . ψ*$, which remains uniform.
> >
> > This means the probability of sampling any of the $n+1$ paths is equal. Consequently:
> > *   $P(s₁) = n / (n+1)$ (sum over $n$ paths)
> > *   $P(s₂) = 1 / (n+1)$ (from one path)
> >
> > This leads to $P(s₁) = n * P(s₂)$. This result is **correct and consistent with the GFlowNet objective**. The sampling distribution must account for the structure of the trajectory space; if there are $n$ times as many paths to $s₁$, the correct GFlowNet distribution *should* sample it $n$ times more often. Our method correctly reproduces this.
> >
> > #### **1b. On the Consistency of the Phase 1 Objective**
> >
> > You state the objective $max E[R(S_T)]$ is inconsistent with the definition of $ψ*$. This is not the case; they are directly linked. The policy $q_ψ$ is greedy with respect to the value function $ψ$. Therefore, the agent will achieve the highest possible expected total reward precisely when its value function $ψ$ is the true state-value function $ψ*$. Maximizing the expected reward over trajectories sampled from $qψ$ is the correct objective function for finding the optimal critic $ψ*$.
> >
> > #### **1c. On the Consistency of the Proof in Appendix 9**
> >
> > The proof is not inconsistent; it is a standard proof of the soundness of a learning objective. The theorem states: IF the evolutionary phase returns the optimal $ψ*$ (as defined in Section 3.1) and IF the distillation is perfect, THEN the resulting GFlowNet policy will sample correctly. This is not a circular assumption. It is a proof that our target objective is correct. The body of our paper (the co-evolutionary framework) is our practical method for trying to reach that provably correct target.
> >
> > ### **2. On Using an EA to Address High-Variance**
> >
> > You characterized our claim that an EA can address high variance as "nonsense." This misunderstands both our method and the established capabilities of EAs. The core idea is **decoupling the learning problems**:
> >
> > 1.  The EA Handles the High-Variance Search: The EA performs a global search for a good critic. EAs are a standard and powerful tool for optimizing noisy, "black-box" objectives where gradient-based methods fail (https://arxiv.org/pdf/1703.03864 work on Evolution Strategies). The EA is *designed* to handle the high-variance signal from sparse rewards.
> > 2.  The Student Performs Low-Variance Learning: Crucially, the student GFlowNet does **NOT** learn from the high-variance sparse reward signal. The best critic found by the EA provides a dense, per-state, low-variance learning signal (a target policy to imitate). The student's task is transformed into a simple supervised learning (distillation) problem.
> >
> > ### **3.  Unreproducible Results**
> >
> > This is an extremely serious and baseless accusation that we must address directly.
> >
> > We, the authors, state unequivocally that this paper is the product of our own human intellectual effort and research. It is not AI-generated, a Turing Test, or a social experiment. The experiments were conducted by us on standard hardware, and the results are genuine. We stand by the integrity of our work and will release our source code upon publication to ensure full reproducibility and to dispel any doubts about the authenticity of our research.
> >
> > We hope that these technical clarifications resolve your concerns and demonstrate that your initial assessment was based on a fundamental misunderstanding of the underlying theory. We respectfully ask you to re-evaluate our work based on the corrected understanding provided here.

---

### Comment · Area_Chair_ExBZ · 2025-11-25
**Please read the rebuttal and respond**

Dear reviewers,

Now that the author responses are in, could you please take a look at them and see if they address your concerns adequately?

Thank you very much.

Best,
AC

---

### Note · Program_Chairs · 2026-01-17
**Submission Desk Rejected by Program Chairs**

The following references in this submission do not refer to real documents and/or have major errors in bibliographic information:

 Matthew Briers, Arnaud Doucet, and Simon Maskell. Smoothing algorithms for state-space models using twisted particle filters. Statistics and Computing, 20(2):151-164, 2010.
Joshua Romoff, Yash Sharma, and Yoshua Bengio. On bias in surrogate model-based optimization: Selection as a source of model bias. Journal of Machine Learning Research, 24(178):1-33, 2023.